# Mechanistic models of asymmetric hand-over-hand translocation and nucleosome navigation by CMG helicase

Fritz Nagae[1,2,3], Yutaka Murata[1], Masataka Yamauchi [1], Shoji Takada [1] & Tsuyoshi Terakawa [1] ✉

Faithful replication of eukaryotic chromatin requires the CMG helicase to translocate directionally along single-stranded DNA (ssDNA) while unwinding double-stranded DNA (dsDNA) and navigating nucleosomes. However, the mechanism by which CMG achieves processive translocation and deals with nucleosomal barriers remains incompletely understood. Here, using coarse-grained molecular dynamics simulations with ATP-driven conformational switching, we show that asymmetric rotational transitions among four distinct ssDNA-binding states enable CMG to achieve directional translocation and DNA unwinding. We further demonstrate that the fork protection complex (Csm3/Tof1) and RPA enhance processivity through distinct mechanisms: Csm3/Tof1 grips the parental duplex to suppress backtracking, while RPA alleviates lagging-strand clogging. Upon nucleosome encounter, Csm3/Tof1 promoted partial unwrapping of the entry DNA, but further progression is energetically restricted near the nucleosomal dyad. The histone chaperone FACT lowers this barrier and simultaneously prevents inappropriate histone transfer to the lagging strand. Our results provide mechanistic insights into how the eukaryotic replisome coordinates helicase activity, nucleosome navigation, histone chaperone function, and histone recycling during eukaryotic DNA replication.

Faithful DNA replication is essential for the accurate transmission of genetic information prior to cell division[1,2]. In eukaryotes, this process is driven by the Cdc45–Mcm2-7–GINS (CMG) replicative helicase, which unwinds double-stranded DNA (dsDNA) into two single-stranded DNA (ssDNA) templates for DNA polymerases[1,2]. To accomplish the directional unwinding, CMG translocates along the leading-strand template in the 3′ → 5′ direction, while sterically excluding the lagging-strand template from its central channel[3].

The Mcm2–7 complex serves as the helicase core of CMG and is composed of a hetero-hexameric ring of AAA+ ATPase subunits that convert ATP hydrolysis into mechanical work to drive unidirectional movement along ssDNA[4–7]. Hexameric AAA+ helicases share a conserved architecture across diverse organisms, including viruses, bacteria, archaea, and eukaryotes, where ATP binding and hydrolysis occur at subunit interfaces[4–7]. Structural, biochemical, and single-molecule studies have established that ATP binding promotes engagement with ssDNA, whereas ATP hydrolysis and subsequent ADP binding trigger disengagement[8–14]. In homo-hexameric AAA+ helicases such as papillomavirus E1[9], bacteriophage T7 gp4[10], SV40 large T-antigen[15], Escherichia coli DnaB[11], and archaeal MCM[8], ATP hydrolysis proceeds sequentially around the ring, facilitating translocation through a symmetric "hand-over-hand" mechanism[4–11]. However,

[1]Department of Biophysics, Graduate School of Science, Kyoto University, Kyoto, Japan. [2]Present address: Structure and Dynamics of Molecular Machines, Max Planck Institute of Biochemistry, Martinsried, Germany. [3]Present address: Department of Bioscience, Technical University of Munich, Garching, Germany. ✉e-mail: terakawa@biophys.kyoto-u.ac.jp

whether the hetero-hexameric Mcm2–7 ring of eukaryotic CMG employs a similar symmetric ATPase cycle remains unclear.

Recent cryo-electron microscopy (cryo-EM) studies have revealed multiple conformations of the eukaryotic CMG complex, each characterized by distinct subsets of Mcm subunits engaging the leading-strand DNA[12,16]. Notably, these structures do not capture all sequential ATP hydrolysis intermediates predicted by the symmetric hand-over-hand model. Complementary biochemical analyses further show that CMG can unwind dsDNA even when ATPase activity is impaired in specific Mcm subunits[12,17]. These observations suggest that a single ATP hydrolysis event may induce coordinated conformational changes across multiple Mcm subunits, leading to rotationally asymmetric shifts in DNA-binding interfaces[12]. While this raises the possibility of an "asymmetric hand-over-hand" translocation mechanism, its physicochemical feasibility and mechanistic details remain unresolved.

Several accessory proteins are known to facilitate CMG-mediated dsDNA unwinding during eukaryotic DNA replication[18–20]. The ssDNA-binding protein RPA primarily binds the lagging-strand ssDNA[21] and has been shown by single-molecule imaging to enhance CMG-driven unwinding[19]. In parallel, the fork protection complex (FPC), composed of Mrc1, Csm3, and Tof1, directly interacts with CMG and is required for rapid and efficient DNA replication[18,20]. Recent cryo-EM studies further revealed that Csm3 and Tof1 engage the parental dsDNA ahead of the replication fork[20]. Despite these insights, the molecular mechanisms by which RPA and the FPC promote CMG translocation and DNA unwinding remain poorly understood.

As DNA replication proceeds, the CMG helicase at the leading edge of the replisome inevitably encounters nucleosomes assembled along the parental DNA[22,23]. Each nucleosome is composed of an H3/H4 tetramer and two H2A/H2B dimers, around which 147 base pairs of DNA are wrapped[24]. To ensure faithful replication, the replisome cooperates with histone chaperones to disassemble and reassemble nucleosomes[22,23]. Importantly, both nucleosome positioning and histone modifications must be faithfully inherited to preserve gene regulation and epigenetic information[23]. Thus, parental histones must be accurately transferred and redeposited onto daughter strands. Our previous coarse-grained simulations visualized parental histone transfer at a static replication fork, without modeling replisome progression[25,26]. A recent Cryo-EM study has identified structural snapshots of the human replisome advancing into a nucleosome up to SHL(−4)[27]. However, the physical mechanism by which the replisome collides with and traverses nucleosomal barriers during translocation remains unclear. In this study, we address this gap by explicitly simulating CMG translocation and its interactions with nucleosomes.

In this study, we perform coarse-grained molecular dynamics simulations of a 20-protein replisome complex to investigate how CMG translocates along ssDNA, unwinds dsDNA, and navigates nucleosomes. By periodically switching conformational states to mimic ATP-driven structural changes[28], we identify four distinct DNA-binding states whose asymmetric transitions enable directional translocation and dsDNA unwinding via heterogeneous steps, supporting the feasibility of an asymmetric hand-over-hand mechanism. Our simulations also reveal distinct roles of accessory factors: Csm3/Tof1 grips the parental duplex to prevent backtracking, while RPA clears the lagging strand from the CMG pore, both promoting efficient unwinding. Upon nucleosome encounter, Csm3/Tof1 facilitates partial unwrapping, but further progression stalls near the dyad. The histone chaperone FACT alleviates this barrier and prevents direct histone transfer to the lagging strand. These results provide testable mechanistic hypotheses for how the replisome coordinates helicase activity, nucleosome navigation, histone chaperone function, and histone recycling during eukaryotic DNA replication.

## Results

### Mechanistic exploration of the CMG translocation along ssDNA

To explore the mechanism of CMG translocation along ssDNA, we built coarse-grained models based on cryo-EM structures of the *Saccharomyces cerevisiae* CMG complex. Among the six available configurations (previously labeled as I–VI)[16], we selected three representative conformations (States 1–3, corresponding to cryo-EM States I, III, and V, respectively) that exhibit distinct ssDNA-binding patterns (Fig. 1A). To identify a minimal set of conformational states sufficient to drive translocation, we excluded State VI because it lacks ssDNA engagement, State II due to its high structural similarity to State I (RMSD = 0.5 Å), and State IV because its low resolution (7.3 Å) may compromise accurate modeling of protein−DNA interactions.

A 495-nucleotide ssDNA was modeled and bound to the CMG to reproduce the protein−DNA contacts observed in the cryo-EM structures (Fig. 1B). We employed the AICG2+ model[29] to describe the protein, representing each amino acid as a single bead placed at the $C_\alpha$ atom position, thereby stabilizing the reference structures while capturing protein fluctuations. The DNA was modeled using the 3SPN2.C model[30], in which each nucleotide is represented by three beads (base, sugar, and phosphate) to preserve the B-form structure and accurately reproduce sequence-dependent curvature, persistence lengths, and melting temperature.

To mimic ATP-driven conformational changes, we implemented a potential-switching scheme, where energy functions stabilizing the three reference *S. cerevisiae* CMG structures were periodically switched during the simulations[28]. Protein−DNA interactions were modeled as hydrogen bonds that switched synchronously with the structure-stabilizing potential, while electrostatic and excluded-volume interactions remained unchanged throughout. In each state, four Mcm subunits bind the ssDNA, forming a right-handed spiral within the central channel of the Mcm2–7 ring (Fig. 1A). The identity of these DNA-engaging subunits varies depending on the state: Mcm2/3/5/6 in State1, Mcm2/3/5/7 in State2, and Mcm3/4/6/7 in State3. Transitioning from State1 to State2 involves release of the 3′ region of ssDNA from Mcm6 and its capture by Mcm7 (Fig. 1C). The State2 → 3 transition proceeds similarly, with dissociation from Mcm2/5 and reassociation with Mcm4/6. Finally, the transition back to State1 entails dissociation from Mcm3/4/7 and binding to Mcm2/3/5. Notably, each transition requires a distinct number of subunits to engage or disengage the DNA, yielding an asymmetric hand-over-hand mechanism (Fig. 1C). As we demonstrate later, the State3 → 1 transition acts as a kinetic bottleneck that compromises translocation efficiency, prompting us to test whether introducing an additional intermediate state could resolve this limitation in the following section.

We performed simulations of *S. cerevisiae* CMG bound to ssDNA using the conformational switching scheme, conducting 20 independent runs with different random seeds. At each state transition, the underlying potential energy function was abruptly switched, after which the *S. cerevisiae* CMG conformation quickly relaxed and reached equilibrium (Supplementary Fig. 1A–C). In 70% of the trajectories (14/20), CMG translocated 12 ± 1 nucleotides in the 3′ → 5′ direction after a single switching cycle (Fig. 1D, Supplementary Movie 1), far exceeding the displacement expected from passive diffusion without switching (Supplementary Fig. 1D). Because the abrupt conformational switching introduces energy into the system, one might be concerned that the observed 12-nt translocation was driven primarily by this artificial energy input. To address this, we performed control simulations in which only the conformation of CMG was switched, while the coarse-grained hydrogen-bonding potentials responsible for DNA engagement were held fixed. Under these conditions, the final displacement of CMG after one complete cycle was 0 ± 6 nt (Supplementary Fig. 1E), indicating that the directional translocation observed in our original simulations is not attributable to the artificial energy introduced by conformational switching alone. Cryo-EM data indicate that each Mcm

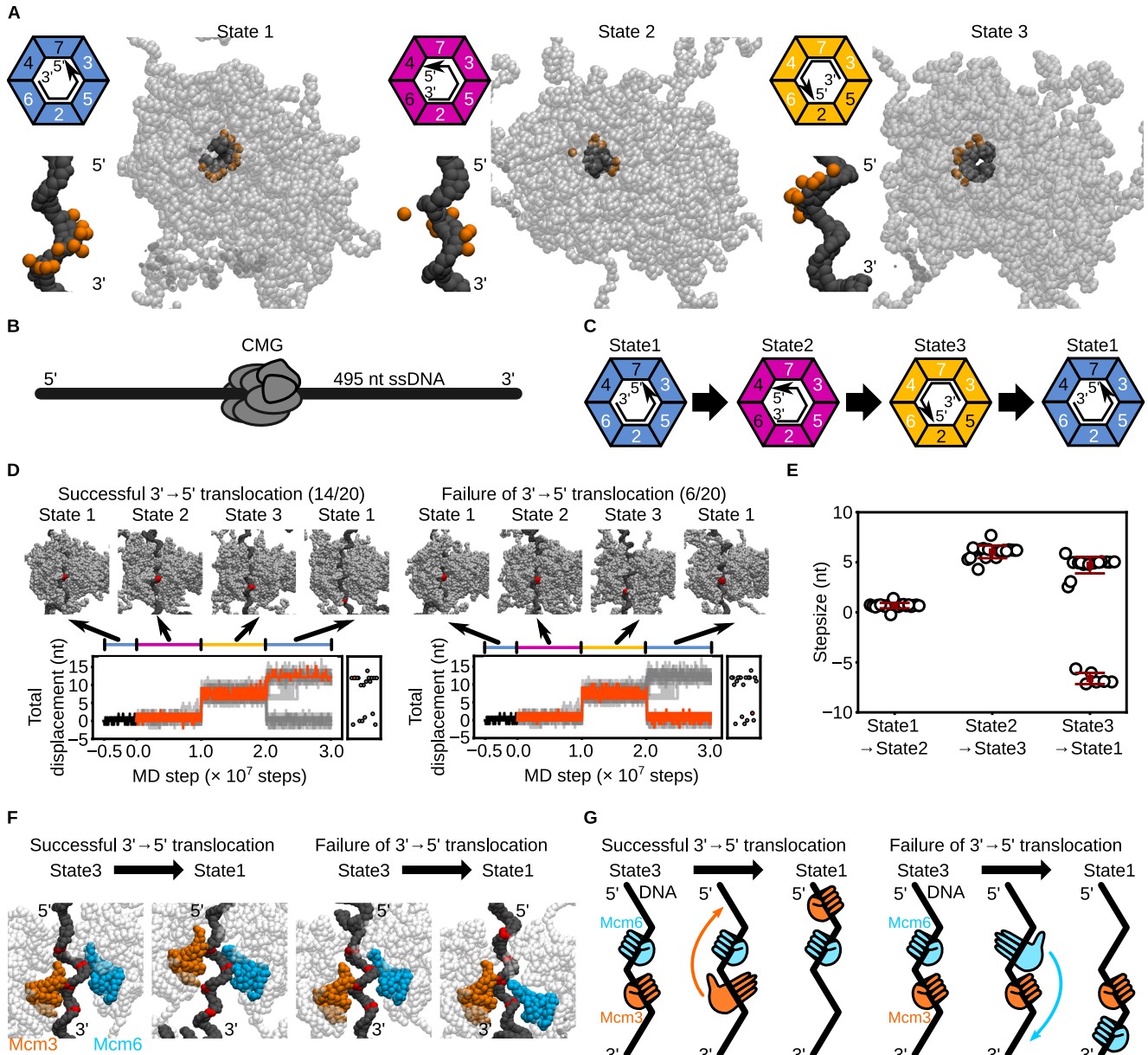

**Fig. 1 | Coarse-grained molecular dynamics simulations of CMG translocation on ssDNA using the potential switching scheme. A** Coarse-grained structures of CMG bound to ssDNA in State1 (left), State2 (middle), and State 3 (right). DNA is shown as black beads; CMG as transparent gray beads. Orange beads indicate Mcm2–7 residues involved in coarse-grained hydrogen-bonding potentials. Colored trapezoids schematically represent the Mcm subunits: blue for State 1, magenta for State 2, and yellow for State 3. Black arrows indicate the 3′ to 5′ direction of ssDNA. White and black numbers in the trapezoids denote subunits with coarse-grained hydrogen-bonding potentials enabled and disabled, respectively. **B** Schematic of the initial model: CMG positioned on a 495-nt ssDNA. DNA and CMG are represented in black and gray, respectively. **C** Sequence of potential switching among States 1, 2, and 3. **D** Representative snapshots and time trajectories of CMG displacement under the switching scheme. Red beads mark the nucleotide initially bound to the ps1β loop of Mcm3. The black trace shows the trajectory from State 1 used for simulations. Orange lines represent forward-translocating (left) and idling (right) trajectories; gray lines represent other cases. Scatter plots at right show final displacements. **E** Step-size per transition for the State1 → 2, 2 → 3, and 3 → 1 transitions. Error bars indicate mean ± standard deviation ($n = 20$, 20, 14, and 6 for State1 → 2, State2 → 3 transitions, State3 → 1 transitions of translocating and idling trajectories). **F** Representative snapshots of CMG in State 3 and State 1 from forward-translocating (left) and idling (right) trajectories. Red beads indicate nucleotides contacted by the ps1β loop of Mcm3 (orange) or Mcm6 (blue) during the transition. **G** Schematic of the State 3 → 1 transition in forward-translocating (left) and idling (right) trajectories, highlighting the subunit anchoring differences. Source data are provided as a Source data file.

subunit interacts with ~2 nt of ssDNA[12,16], so one full cycle involving all six subunits would yield a 12-nt translocation—consistent with our simulations. The individual step sizes for the transitions from State1 → 2, State2 → 3, and State3 → 1 were 1 ± 0.2 nt, 6 ± 0.6 nt, and 5 ± 1 nt, respectively (Fig. 1D, E), revealing a heterogeneous stepping pattern.

In the remaining 30% of trajectories (6/20), translocations from State1 to State2 and from State2 to State3 occurred in the 3′→5′ direction; however, the subsequent State3 → 1 transition resulted in a backward step of −7 ± 0.6 nt, yielding a net displacement of 0 ± 1 nt (Fig. 1D, E, Supplementary Movie 2). This backward step is likely caused by incomplete coordination during the State3→State1 transition, which requires disengagement of the 3′ region from Mcm3/7/4 and reassociation of the 5′ region with Mcm2/3/5 (Fig. 1C). Notably, both Mcm3 and Mcm6 interact with ssDNA in State3 and State1. Depending on

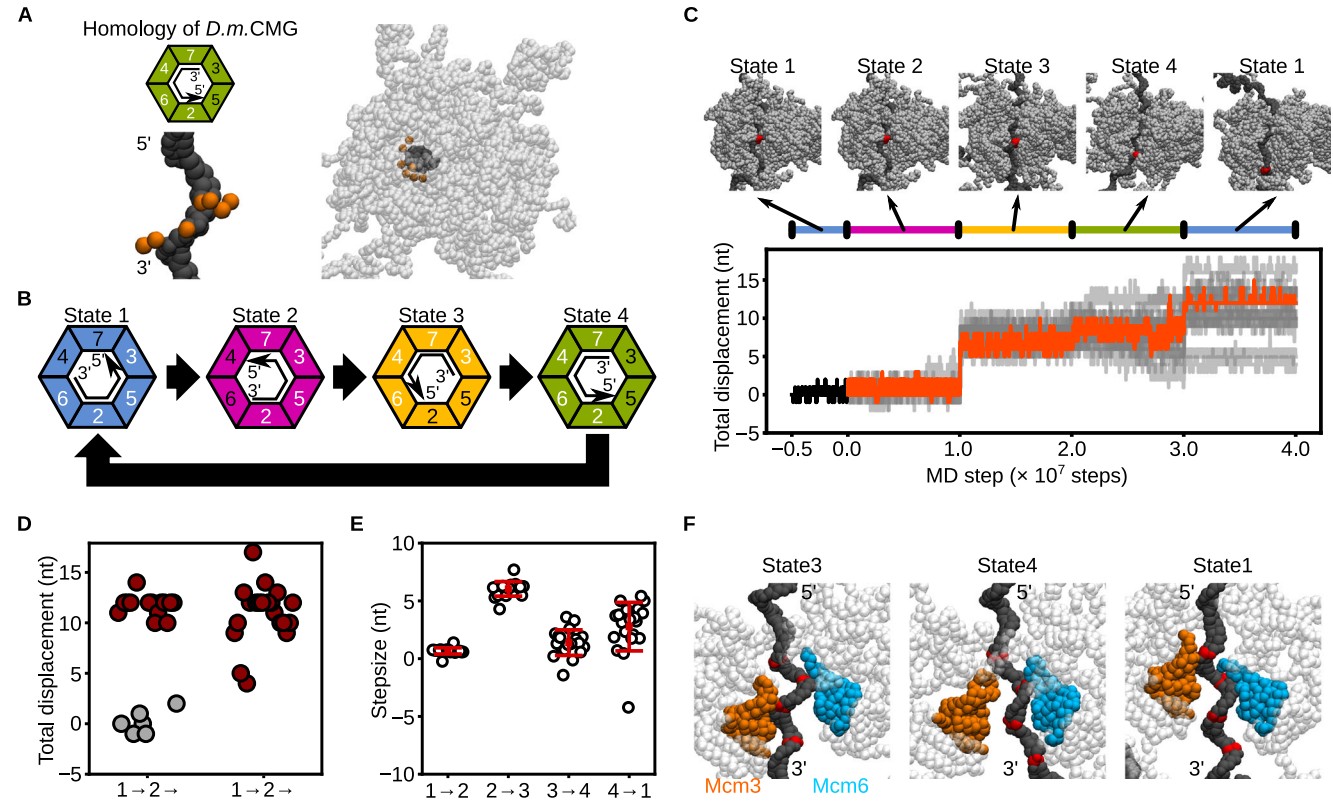

**Fig. 2 | The four-state asymmetric hand-over-hand model of CMG translocation. A** Coarse-grained structure of *S. cerevisiae* CMG bound to ssDNA, modeled by homology to *Drosophila melanogaster* CMG. DNA and CMG are shown as black and transparent gray beads, respectively. Orange beads indicate residues in Mcm2–7 where coarse-grained hydrogen-bonding potentials were applied. Green trapezoids schematically represent the Mcm subunits in State 4. Black arrows indicate the direction of the ssDNA. White and black numbers in the trapezoids denote subunits with coarse-grained hydrogen-bonding potentials enabled or disabled, respectively. **B** Schematic illustration of the conformational switching sequence: State1 → 2 → 3 → 4 → 1. **C** Representative snapshots and time trajectories of CMG displacement under the four-state switching scheme. Red beads indicate the nucleotide initially bound to the ps1β loop of Mcm3. The black line represents a trajectory starting from State 1. The orange line shows a forward-translocating trajectory; gray lines represent all other trajectories. **D** Final displacements from simulations with and without inclusion of State4. Red and gray circles represent trajectories that showed forward translocation or idling, respectively. **E** Step-size per transition for the State1 → 2, 2 → 3, 3 → 4, 4 → 1 transitions. Error bars represent mean ± standard deviation (n = 20 for all transitions). **F** Representative snapshots of CMG in States 3, 4, and 1. Color scheme is consistent with Fig. 1F. Source data are provided as a Source data file.

which subunit remains bound during the State3 → 1 transition, the outcome of the conformational change differs. When Mcm6 remains bound during the transition, the ps1β loop of Mcm3 captures the 5′ region, promoting forward 3′ → 5′ translocation (Fig. 1F, G). Conversely, if Mcm3 remains bound, the ps1β loop of Mcm6 binds to the 3′ region after the transition, leading to backward 5′ → 3′ movement. The anchoring subunit—Mcm3 or Mcm6—is stochastically determined in each trajectory, leading to variability in translocation direction. These results suggest that, despite an overall asymmetry favoring forward movement, subunit-level miscoordination during the State3 → 1 transition can cause idling or reversal.

We next considered an alternative pathway of the conformational transitions, State1 → 3 → 2 → 1. Successful 3′ → 5′ translocation would require complete dissociation of the ssDNA from *S. cerevisiae* CMG during the State3 → 2 transition and its subsequent rebinding during the State2 → 1 transition (Supplementary Fig. 1F). To assess the feasibility of this pathway, we performed 20 simulations under the State1 → 3 → 2 → 1 switching scheme. However, no 3′ → 5′ translocation was observed in any trajectory (0/20). Instead, 5% (1/20) of trajectories exhibited a 12-nt backward movement, while the remaining 95% (19/ 20) showed idling behavior (Supplementary Fig. 1F). These results indicate that the sequential pathway State1 → 2 → 3 → 1 is necessary for productive translocation along ssDNA.

A separate cryo-EM study identified four distinct conformations of *Drosophila melanogaster* CMG[12], in which different combinations of Mcm subunits engage ssDNA. Notably, the *D. melanogaster* CMG State2A in the original literature[12], where Mcm7/4/6/2 bind the ssDNA (Fig. 2A), resembles a partially characterized conformation of *S. cerevisiae* CMG reported previously[20]. We hypothesized that incorporating an analogous conformation as an additional intermediate (State4) could subdivide the challenging State3 → 1 transition into two sequential steps—first involving Mcm2, followed by Mcm3/5 engagement (Fig. 2B)—thereby reducing the likelihood of backward steps. To test this, we constructed *S. cerevisiae* CMG State4 via homology modeling using *D. melanogaster* CMG State2A as a template (Fig. 2A) and performed 20 simulations with the modified switching cycle State1 → 2 → 3 → 4 → 1 (Fig. 2B, Supplementary Movie 3). At the transition to State4, the underlying potential energy function was abruptly switched, after which the *S. cerevisiae* CMG conformation quickly relaxed and reached equilibrium (Supplementary Fig. 1G). Strikingly, all the trajectories (20/20) exhibited net forward translocation along ssDNA in the 3′ → 5′ direction (Fig. 2C, D), with 18 out of 20 trajectories achieving displacements close to the expected 12 nucleotides per cycle (Fig. 2D). The observed step sizes were 1 ± 0.2 nt (1 → 2), 6 ± 0.6 nt (2 → 3), 2 ± 1 nt (3 → 4), and 3 ± 1 nt (4 → 1), yielding a total displacement of 11 ± 3 nt (Fig. 2C–E).

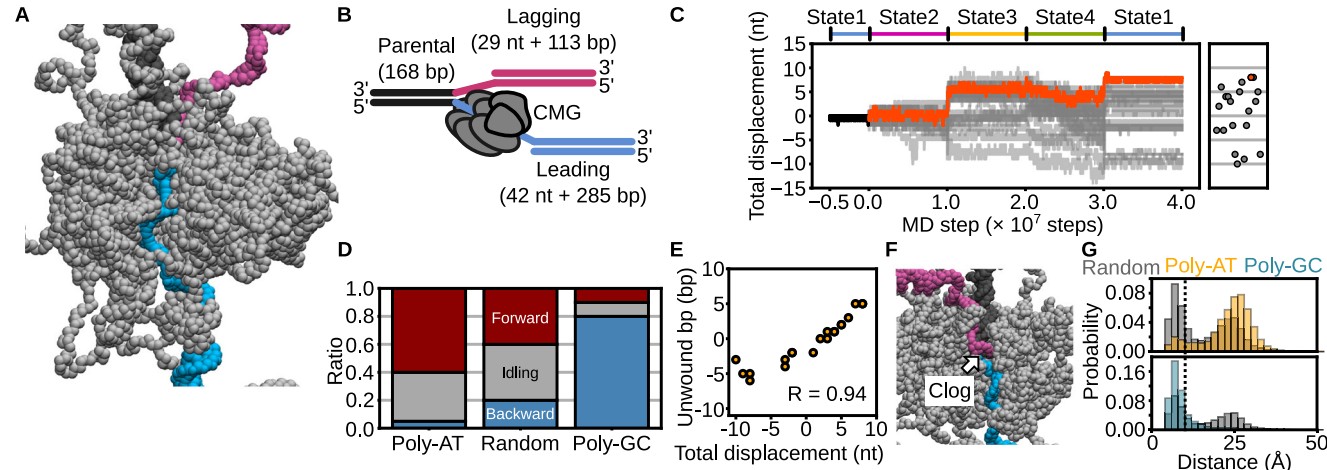

**Fig. 3 | Coarse-grained molecular dynamics simulations of dsDNA unwinding by CMG using the four-state asymmetric hand-over-hand model. A, B** Initial model of CMG bound to Y-forked DNA. **A** Coarse-grained representation; **B** schematic diagram. DNA strands are colored as follows: parental (black), leading (cyan), lagging (magenta); CMG is shown in gray. Mcm7 is omitted to visualize the path of the leading-strand ssDNA through the Mcm2–7 ring. **C** Time trajectories of CMG displacement under the four-state switching scheme. Red beads indicate the nucleotide initially bound to the ps1β loop of Mcm3. The black trace represents the trajectory used for simulation initiation. The orange line shows a representative forward-translocating trajectory; gray lines indicate all others. Final displacements are summarized in the scatter plot. **D** Classification of translocation outcomes based on final displacement: forward (> +3 bp, red), backward (<−3 bp, blue), and idling (−3 to +3 bp, gray) for each DNA sequence condition (poly-AT, random, poly-GC). **E** Correlation between total displacement and number of base pairs unwound at the final frame (*R* = 0.94). **F** Representative snapshot showing clogging of the lagging strand within the central pore of CMG. **G** Distribution of the minimum distance between lagging-strand beads and CMG pore residues. The dashed line at 10 Å indicates the threshold used to define clogging. Source data are provided as a Source data file.

Structural analysis revealed how the inclusion of State4 facilitates forward translocation. During the State3 → 4 transition ($1-2 \times 10^7$ MD steps to $2-3 \times 10^7$ MD steps), Mcm3 released the 3′ region of the ssDNA, while Mcm6 anchored the 5′ region (Fig. 2F). In the subsequent State4 → 1 transition ($3-4 \times 10^7$ MD steps), Mcm3 re-engaged the 5′ region, with Mcm6 still anchored. This sequential engagement ensured that Mcm6, rather than Mcm3, served as the anchor during the critical transition, thereby biasing the system toward productive 3′ → 5′ translocation.

A recent cryo-EM study proposed that *S. cerevisiae* CMG translocation could be driven by a planar-to-spiral transition of DNA-binding loops, in which the planar and spiral configurations correspond to our State1 and State4 structures, respectively[31]. To test whether transitions between two states are sufficient, we systematically performed 20 simulations for all possible two-state switching schemes among State1–4 (Supplementary Fig. 2A–F). Notably, the State1 → 4 → 1 scheme, corresponding to the proposed planar-to-spiral transition, resulted in forward translocation in only 4/20 trajectories (20%) (Supplementary Fig. 2C). Other two-state transitions, such as State1 → 3 → 1, State2 → 3 → 2, State2 → 4 → 2, and State3 → 4 → 3, also showed lower translocation frequencies compared to the full four-state cycle (Supplementary Fig. 2B, 2D–H). These results highlight that transitions involving at least four distinct conformational states enable more robust 3′ → 5′ translocation than simpler two-state models.

Collectively, our simulations suggest that, while the available cryo-EM structures of *S. cerevisiae* CMG[16] have provided critical insights, they may not fully capture all the conformational intermediates required for directional translocation on ssDNA. Our results support the existence of an additional intermediate (State4), which may act as a transient state, and indicate that at least four distinct states are involved in the translocation mechanism (Fig. 2B). Furthermore, this suggests that as-yet unresolved conformational states may also contribute to stabilizing and smoothing the transitions between DNA-bound states. Therefore, our findings do not definitively exclude the possibility of a symmetric hand-over-hand mechanism. Nonetheless, previous biochemical evidence demonstrating that not all

Mcm2–7 ATPase subunits are essential for translocation[12,17] supports the plausibility of an asymmetric hand-over-hand model.

## Mechanistic exploration of the dsDNA unwinding by CMG

We next investigated whether the State1 → 2 → 3 → 4 → 1 conformational switching scheme could reproduce *S. cerevisiae* CMG-driven unwinding of dsDNA. To this end, we constructed a Y-forked DNA substrate containing 168, 113, and 285 bp of dsDNA for the parental, lagging, and leading strands, respectively, along with 29 nt and 42 nt of ssDNA for the lagging and leading strands (Fig. 3A, B). We then performed 20 simulations of *S. cerevisiae* CMG bound to the Y-forked DNA under this switching scheme (Supplementary Movie 4).

Among the 20 trajectories, 40% (8/20) exhibited forward translocation of 4–8 nucleotides, whereas the remaining 60% (12/20) showed slippage or backward steps during some transitions (Fig. 3C, D). The maximum displacement was limited to 8 nt, smaller than the ideal 12 nt per cycle (Fig. 3C). By classifying trajectories with a net displacement between −3 and +3 nt as idling, and those with −4 nt or less as backward-stepping, we found that, under the random DNA sequence condition, 40% (8/20) were idled and 20% (4/20) showed backward steps (Fig. 3D, random). These results suggest that base pairs positioned ahead of *S. cerevisiae* CMG impede its translocation along ssDNA. Notably, trajectories that achieved forward translocation displayed a one-to-one coupling between the number of unwound base pairs and the number of translocated nucleotides (Fig. 3E), indicating that our simulation approach captures tightly coupled dsDNA unwinding and translocation.

We hypothesized that if base pairs immediately upstream of *S. cerevisiae* CMG serve as an energetic barrier, the efficiency of translocation would depend on the DNA sequence. To test this, we performed an additional 20 simulations using Y-forked DNA with either poly-AT or poly-GC sequences replacing the random sequence. Translocation of 4 nt or more occurred in 60% (12/20) of trajectories with poly-AT DNA, but only in 10% (2/20) with poly-GC DNA (Fig. 3D). Since GC base pairs are energetically more stable than AT pairs[32], these results support the idea that upstream base pairs impose an energy barrier to CMG progression. Notably, regardless of sequence, the

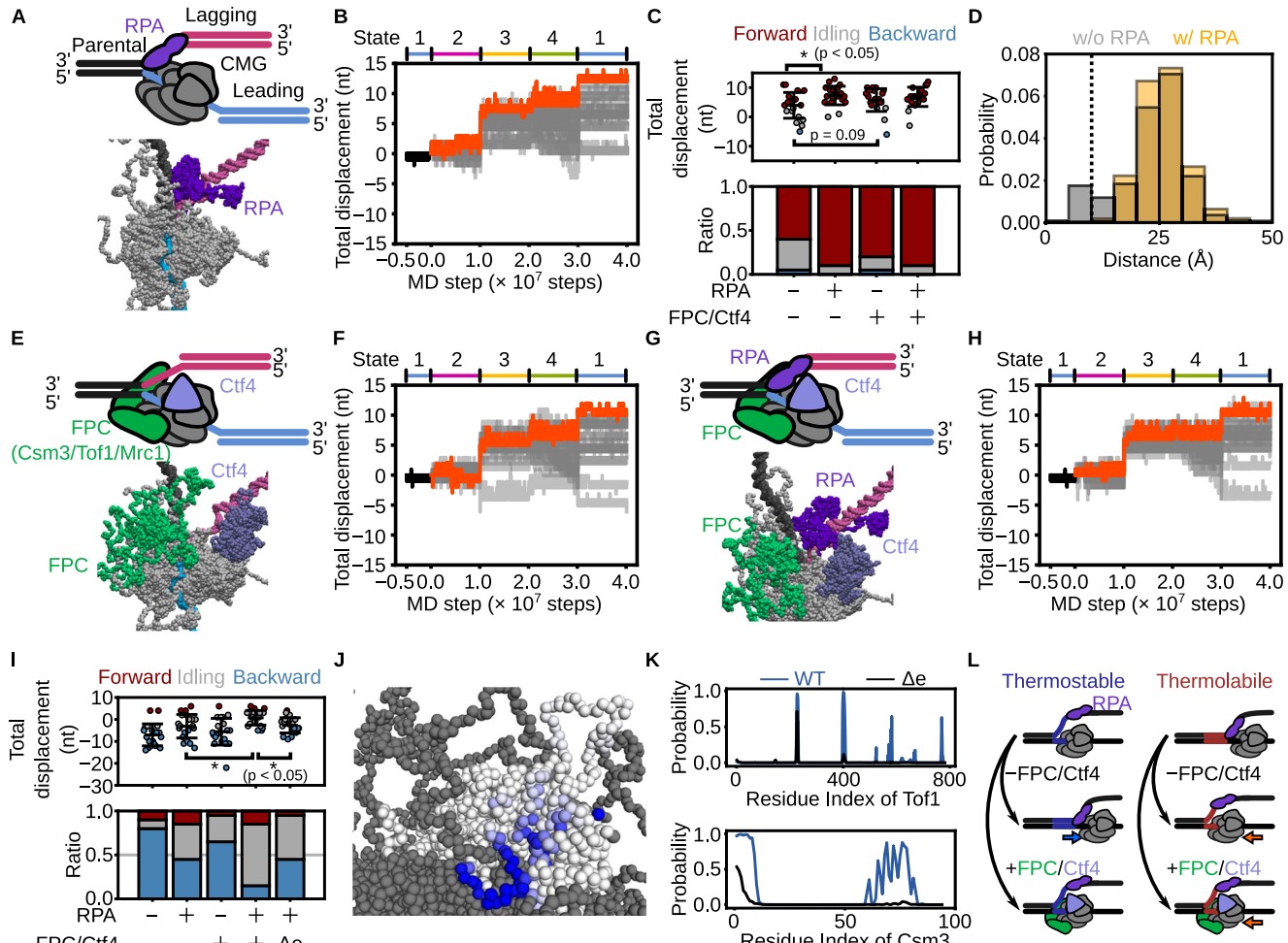

**Fig. 4 | Coarse-grained molecular dynamics simulations of dsDNA unwinding by CMG in the presence of RPA, the fork protection complex (FPC: Mrc1/Tof1/Csm3), and Ctf4. A, E, G** Initial configurations and schematic representations of the Y-forked DNA substrate bound to CMG with: **A** RPA, **E** FPC and Ctf4, and **G** RPA, FPC, and Ctf4. RPA, FPC, and Ctf4 are shown in purple, green, and light blue, respectively. **B, F, H** Time trajectories of total CMG displacement under each condition. The black line indicates the trajectory from State1 used for simulation initiation. Orange lines represent forward-translocating trajectories; gray lines represent all others. **C** Final displacements (top) and proportions of forward (> +3 bp, red), backward (<−3 bp, blue), and idling (−3 to +3 bp, gray) trajectories (bottom). *P* values were calculated using the one-tailed Wilcoxon–Mann–Whitney test [*P* = 0.006 (Cohen's d = 0.87) between without and with RPA]. Error bars indicate mean ± standard deviation (*n* = 20 for all condition*s*). **D** Distribution of the minimum distance between lagging-strand beads and residues forming the CMG

central pore, with and without RPA. The dashed line at 10 Å indicates the threshold used to define clogging. **I** Final displacements (top) and trajectory classification (bottom) under poly-GC sequences with various combinations of RPA, FPC, and Ctf4. *P* values were calculated using the one-tailed Wilcoxon–Mann–Whitney test [*P* = 0.01 (Cohen's *d* = 0.86) between without and with FPC/Ctf4, and *P* = 0.003 (Cohen's *d* = 1.02) between the presence and absence of electrostatic interactions]. Error bars indicate mean ± standard deviation (*n* = 20 for all condition*s*). **J, K** Probability of each residue in Tof1 and Csm3 contacting the parental strand. **J** Shades of blue in the structural model reflect contact frequency. **K** Blue and black lines represent simulations with and without electrostatic interactions, respectively. **L** Schematic illustration summarizing the roles of RPA and Tof1/Csm3 in facilitating CMG-driven unwinding of thermolabile (poly-AT) and thermostable (poly-GC) DNA sequences. Source data are provided as a Source data file.

number of nucleotides translocated consistently matched the number of base pairs unwound (Supplementary Fig. 3A), indicating that the coupling between translocation and unwinding is a general feature of the mechanism.

We further observed that the reannealed dsDNA/ssDNA junction often clogs the central pore of *S. cerevisiae* CMG during translocation (Fig. 3F, Supplementary Fig. 3B). Reannealing at the dsDNA/ssDNA junction was observed more frequently in simulations with the poly-GC sequence than with the poly-AT sequence (Supplementary Fig. 3A). Furthermore, defining clogging as a minimum distance of less than 10 Å between the lagging strand and the central pore, we found that its occurrence was most frequent with poly-GC sequences, followed by random and then poly-AT sequences (Fig. 3G). These results suggest that stronger base-pairing stability exacerbates clogging of the lagging strand, which in turn hinders CMG translocation.

Since 40% (8/20) of the trajectories using Y-fork DNA with poly-AT sequences exhibited slippage or backward stepping, we hypothesized that additional protein factors may stabilize forward translocation by preventing backward stepping. As a first candidate, we examined the ssDNA-binding protein RPA, previously shown to facilitate CMG-driven dsDNA unwinding[19,33]. We performed 20 simulations in which RPA was bound to the lagging-strand ssDNA while *S. cerevisiae* CMG engaged the leading-strand ssDNA (Fig. 4A, B, Supplementary Movie 5). In the presence of RPA, 90% (18/20) of the trajectories exhibited forward translocation of at least 4 nucleotides (Fig. 4C), significantly improving mean displacement compared to the RPA-free condition (*p* < 0.05, Wilcoxon–Mann–Whitney (WMW) test; Fig. 4C). Moreover, RPA reduced the frequency of clogging events (Fig. 4D). Similar or even stronger effects were observed when using poly-GC sequences (Supplementary Fig. 4A, B). These findings suggest that RPA enhances

dsDNA unwinding by preventing lagging-strand ssDNA from entering the CMG central pore, thereby promoting more efficient and processive translocation.

To further test whether clogging of the central pore by lagging-strand ssDNA inhibits CMG translocation, we replaced the 29-nt single-stranded region of the lagging strand with duplex DNA and performed 20 simulations of unwinding poly-GC sequences (Supplementary Fig. 4C, D). Compared to the condition with exposed ssDNA, this duplex configuration resulted in significantly greater final displacements ($p < 0.05$, Wilcoxon–Mann–Whitney test; Supplementary Fig. 4E). Moreover, the lagging-strand DNA was more effectively excluded from the CMG pore when occluded by duplex formation than when exposed as ssDNA (Supplementary Fig. 4F). Together, these results indicate that preventing lagging-strand clogging—either through duplex formation or RPA binding—enhances CMG translocation, consistent with findings from a previous study[19].

We next tested whether the fork-protection complex (FPC; composed of Mrc1, Tof1, and Csm3) and Ctf4 contribute to stabilizing CMG translocation. These factors interact with CMG and form a scaffold that recruits additional replication-associated proteins[18,20,34–37]. We performed 20 simulations using poly-AT Y-forked DNA, in which *S. cerevisiae* CMG was assembled with FPC and Ctf4 in the absence of RPA (Fig. 4E, F, Supplementary Movie 6). Although 80% (16/20) of trajectories exhibited forward translocation of at least 4 nucleotides in the presence of FPC and Ctf4, the overall displacement was not significantly different from the condition lacking these factors ($P = 0.09$, Wilcoxon–Mann–Whitney test; Fig. 4C). Furthermore, when RPA was combined with FPC and Ctf4 (Fig. 4G, H; Supplementary Movie 7), forward translocation of ≥4 nt was observed in 90% (18/20) of trajectories, identical to the RPA-only condition (90%; 18/20; Fig. 4C). These results indicate that, on poly-AT DNA, the addition of Csm3/Tof1 does not significantly enhance forward translocation beyond the effect of RPA alone.

In contrast, on poly-GC DNA—where translocation is generally inefficient—FPC and Ctf4 markedly suppressed backward stepping. Backward steps occurred in 45% (9/20) of trajectories with RPA alone, 65% (13/20) with FPC and Ctf4 alone, and only 15% (3/20) when all three factors were present (Fig. 4I). Moreover, under the FPC + Ctf4 + RPA condition, the final displacement per cycle was significantly greater than in the RPA-only condition ($P < 0.05$, Wilcoxon–Mann–Whitney test; Fig. 4I). During translocation, Csm3/Tof1 directly engaged the parental dsDNA, consistent with previous cryo-EM observations[20], whereas Mrc1 and Ctf4 exhibited minimal interaction with DNA (Fig. 4J, K, and Supplementary Fig. 5). This binding appeared to be mediated by electrostatic interactions between positively charged residues on Csm3/Tof1 and the DNA backbone (Supplementary Fig. 4G). Disabling these electrostatic interactions reduced DNA-contact probabilities (Fig. 4K), increased the frequency of backward steps to 45% (9/20), and significantly reduced final displacement ($P < 0.05$, Fig. 4I). These findings indicate that anchoring Csm3/Tof1 to the parental duplex is critical for suppressing backtracking on thermostable sequences. In contrast, on thermolabile sequences such as poly-AT, where backward stepping is rare, the effect of Csm3/Tof1 is largely masked (Fig. 4L).

## Mechanistic exploration of *S. cerevisiae* CMG translocation through nucleosomes

In vivo, *S. cerevisiae* CMG functions within chromatin and must translocate along DNA packaged into nucleosomes. To assess the impact of nucleosomes on *S. cerevisiae* CMG activity, we performed 20 simulations in which *S. cerevisiae* CMG—assembled with FPC and Ctf4—was positioned on the leading-strand ssDNA of a Y-forked DNA substrate, with a nucleosome assembled on the parental strand using the Widom 601 positioning sequence. The leading edge of the *S. cerevisiae* CMG–FPC complex was positioned at SHL(−7) on the nucleosomal DNA, placing the fork junction approximately 22 base pairs upstream,

and RPA was bound to the lagging strand (Fig. 5A, B; Supplementary Movie 8). In these simulations, 60% (12/20) of trajectories exhibited translocation of ≥4 base pairs, tightly coupled to unwinding (Fig. 5C–E), a frequency comparable to simulations without nucleosomes (55%; 11/20) (Fig. 5D, E). These results suggest that nucleosomes positioned ≥22 bp downstream of the fork junction do not impede *S. cerevisiae* CMG translocation or unwinding.

When *S. cerevisiae* CMG–FPC was positioned at SHL(−7), placing the fork junction 22 bp upstream of the nucleosome, we observed partial unwrapping of nucleosomal DNA during translocation, facilitated by interactions between Csm3/Tof1 and the nucleosome (Fig. 5C). At the early stage of the simulation (up to $1.0 \times 10^7$ steps; corresponding to ~1 nt of translocation), the nucleosome remained stably wrapped up to SHL(−6.5), although transient unwrapping and rewrapping events were occasionally observed between SHL(−7) and SHL(−5.5) [Fig. 5C (i), (ii)], consistent with simulations conducted in the absence of *S. cerevisiae* CMG–FPC[38]. As displacement reached 6 nt (up to $3.0 \times 10^7$ steps), unwrapping extended to SHL(−4.5), and became stabilized by 10 nt displacement [Fig. 5C (iii)]. Among the trajectories with ≥5 nt translocation, 42% (5/12) showed unwrapping to SHL(−4.5) [Fig. 5C (iii), 5 F]. These observations suggest that *S. cerevisiae* CMG translocation and nucleosome unwrapping are only weakly coupled. Notably, during the *S. cerevisiae* CMG translocation, Csm3/Tof1 engaged the parental dsDNA in a manner similar to that shown in Fig. 4K, whereas interactions between Mrc1/Ctf4 and DNA, as well as between Mrc1, Csm3/Tof1, Ctf4, RPA, and histones, were rarely observed (Supplementary Figs. 6 and 7). When electrostatic interactions between Csm3/Tof1 and DNA were removed, no unwrapping beyond SHL(−5) was observed (0/20 trajectories; Fig. 5G), indicating that Csm3/Tof1 actively promotes entry-end unwrapping via electrostatic engagement with DNA.

We next investigated whether *S. cerevisiae* CMG can translocate along nucleosomal DNA without encountering an energy barrier. Specifically, we assessed translocation efficiency as the leading edge of *S. cerevisiae*. CMG advanced into the nucleosome up to SHL(−5), SHL(−4), and SHL(−3). To model this, we aligned the Y-fork junction 22 bp upstream of SHL(−5), SHL(−4), or SHL(−3), thereby placing the leading edge of the *S. cerevisiae* CMG–FPC complex directly at these positions without any steric clash within the model (Fig. 6A, B; See "Methods" for details). For each configuration, we performed 20 simulations using the same conformational switching scheme (Fig. 6C–E; Supplementary Movies 9–11). Translocation of ≥4 nt occurred in 40% (8/20) of trajectories from SHL(−5) or SHL(−4), but in only 10% (2/20) from SHL(−3) (Fig. 6F bottom). Mean displacements after one cycle progressively decreased as *S. cerevisiae* CMG advanced deeper into the nucleosome: $4 \pm 4$ nt at SHL(−7), $2 \pm 3$ nt at SHL(−5), $1 \pm 4$ nt at SHL(−4), and $0 \pm 3$ nt at SHL(−3) (Fig. 6F top). These findings indicate that as *S. cerevisiae* CMG approaches the nucleosomal dyad, and translocation becomes increasingly restricted, consistent with the presence of a positional energy barrier.

We then analyzed the trajectories from the previous simulations initiated at SHL(−5), SHL(−4), or SHL(−3) to examine the coupling between *S. cerevisiae* CMG translocation and nucleosomal DNA unwrapping. When starting from SHL(−5), no further unwrapping beyond SHL(−4) was observed in any trajectory (Fig. 6G left). In contrast, unwrapping extended to SHL(−2.5) in 38% (3/8) of the trajectories initiated from SHL(−4) (Fig. 6G middle). Interestingly, this level of unwrapping also occurred in some trajectories that did not exhibit translocation, suggesting that spontaneous unwrapping may facilitate CMG progression. A similar pattern was seen from SHL(−3), where unwrapping to SHL(−2) was observed in all translocating trajectories, and in some non-translocating ones as well (Fig. 6G right). These observations indicate that tight histone–DNA interactions around SHL(−4.5) limit spontaneous unwrapping and act as a significant barrier to helicase progression. This is consistent with previous reports

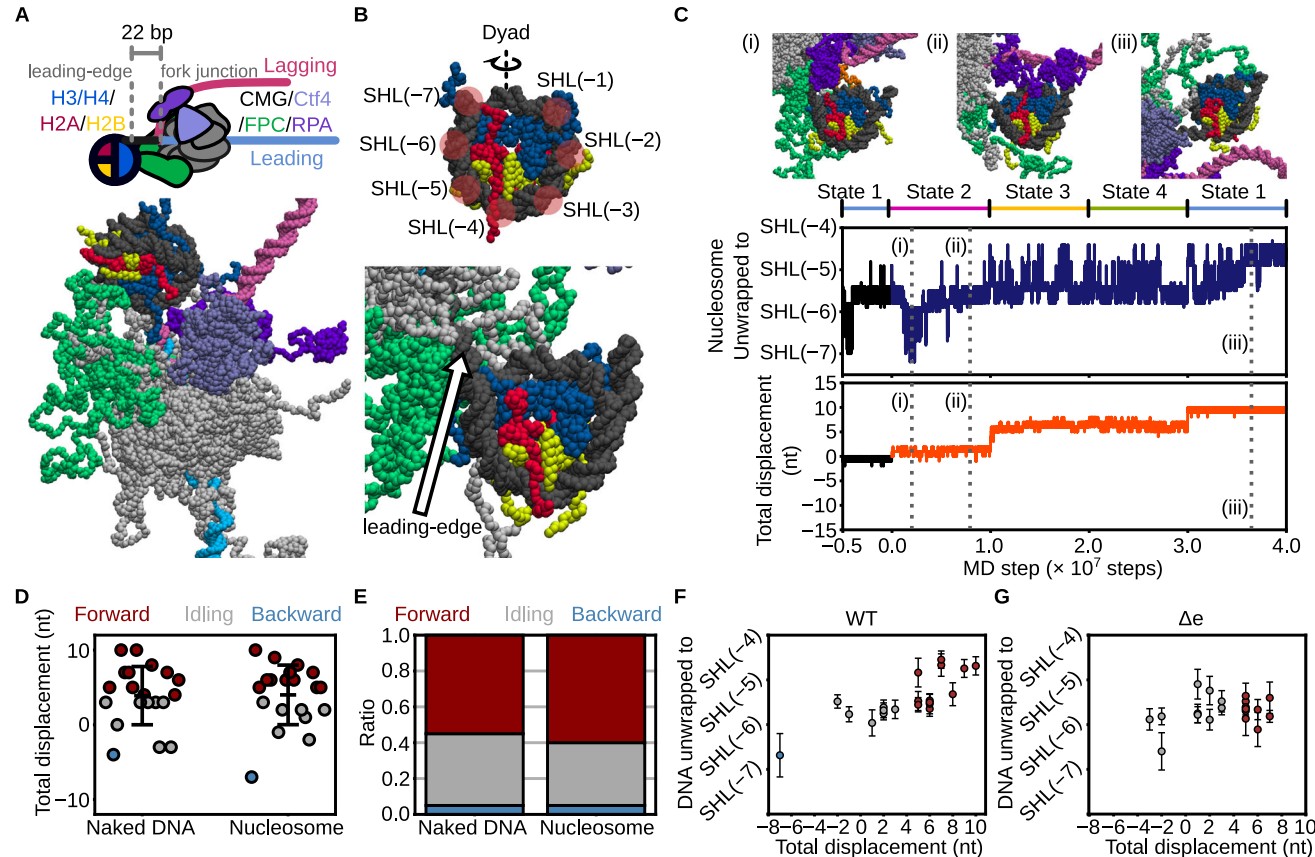

**Fig. 5 | Coarse-grained molecular dynamics simulations of CMG collisions with a nucleosome. A** Schematic (top) and coarse-grained representation (bottom) of the initial configuration of CMG bound to a Y-forked DNA in the presence of RPA, Mrc1, Tof1, Csm3, Ctf4, and a nucleosome. Histones are colored as follows: H3/H4 (blue), H2A (red), and H2B (yellow). **B** Coarse-grained model of a nucleosome core particle with the definitions of superhelical location (SHL) (top) and a nucleosome where a Y-forked DNA engaged CMG collides (bottom). **C** Representative snapshots of nucleosome unwrapping during CMG translocation and time trajectories showing the extent of unwrapping (top) and CMG displacement (bottom). The black line denotes the trajectory from State 1 used for simulation initiation. Blue

and orange lines represent forward-translocating trajectories. **D, E** Final CMG displacements (**D**) and classification of trajectories based on displacement: forward (> +3 bp, red), backward (<–3 bp, blue), and idling (–3 to +3 bp, gray) (**E**), for conditions with and without a nucleosome. Error bars indicate mean ± standard deviation (n = 20 for both conditions). **F, G** Correlation between final CMG displacement and unwrapping extent in simulations with (**F**) and without (**G**) electrostatic interactions between Tof1/Csm3 and DNA. Red, gray, and blue circles indicate forward, idling, and backward trajectories, respectively. Error bars indicate mean ± standard deviation over the final 100 frames (n = 100). Source data are provided as a Source Data file.

showing that SHL(–4.5) represents a key barrier to RNA polymerase II during transcription[39,40] or the replisome during DNA replication[27].

Interestingly, in simulations initiated from SHL(–4) or SHL(–3), we observed that spontaneous unwrapping–occurring even without translocation–exposed the surface of the H2A/H2B dimer, allowing the lagging-strand DNA to bind directly to it (Fig. 6G, H; Supplementary Movie 12). This binding was not observed in simulations starting from SHL(–5), where the H2A/H2B surface remained occluded by the parental strand (Fig. 6G). These results suggest that when *S. cerevisiae* CMG progresses into the nucleosome beyond SHL(–4), partial unwrapping can expose histone surfaces capable of capturing the lagging strand, potentially promoting direct histone transfer and recycling during replication.

## FACT facilitated the CMG translocation and modulates histone recycling

The histone chaperone FACT, a heterodimer of Spt16 and Pob3, is known to facilitate the progression of RNA polymerase II through nucleosomal DNA[39,41]. Structural evidence supports this function: a cryo-EM study revealed FACT bound to a nucleosome unwrapped up to SHL(–4.5) in the presence of RNA polymerase II[39]. FACT has also been shown to be essential for replication through nucleosome arrays[42–44]. Given that our simulations identified a high-energy barrier

to *S. cerevisiae* CMG translocation within the nucleosome, we next sought to elucidate how FACT modulates CMG progression and histone dynamics during chromatin replication.

To assess the role of FACT in *S. cerevisiae* CMG translocation, we introduced FACT into the structural models used in Fig. 6A, B, where the fork junction was positioned 22 base pairs upstream of SHL(–5), SHL(–4), or SHL(–3). FACT was positioned onto the nucleosome dyad based on its location observed in the cryo-EM structure of the RNAP II–nucleosome–FACT complex[39]. We conducted 20 simulations for each condition using the conformational switching scheme to drive translocation (Fig. 7A, B, Supplementary Movie 13). In all cases, the C-terminal intrinsically disordered region (IDR) of Spt16 (residues 949–1035) rapidly bound the H2A/H2B surface exposed by unwrapping, thereby preventing re-association of DNA with that region (Fig. 7C).

When starting from SHL(–5) or SHL(–4), forward translocation of ≥4 nt occurred in 45% (9/20) of trajectories–similar to simulations without FACT (40%; 8/20; Fig. 7D). However, from SHL(–3), FACT increased this proportion to 40% (8/20), compared to 10% (2/20) without FACT (Fig. 7D). Notably, in SHL(–3) simulations, DNA unwrapping extended to SHL(–2) shortly after simulation onset (Fig. 7E). Furthermore, in SHL(–5) simulations, additional unwrapping beyond SHL(–3) occurred in 45% (9/20) of trajectories only

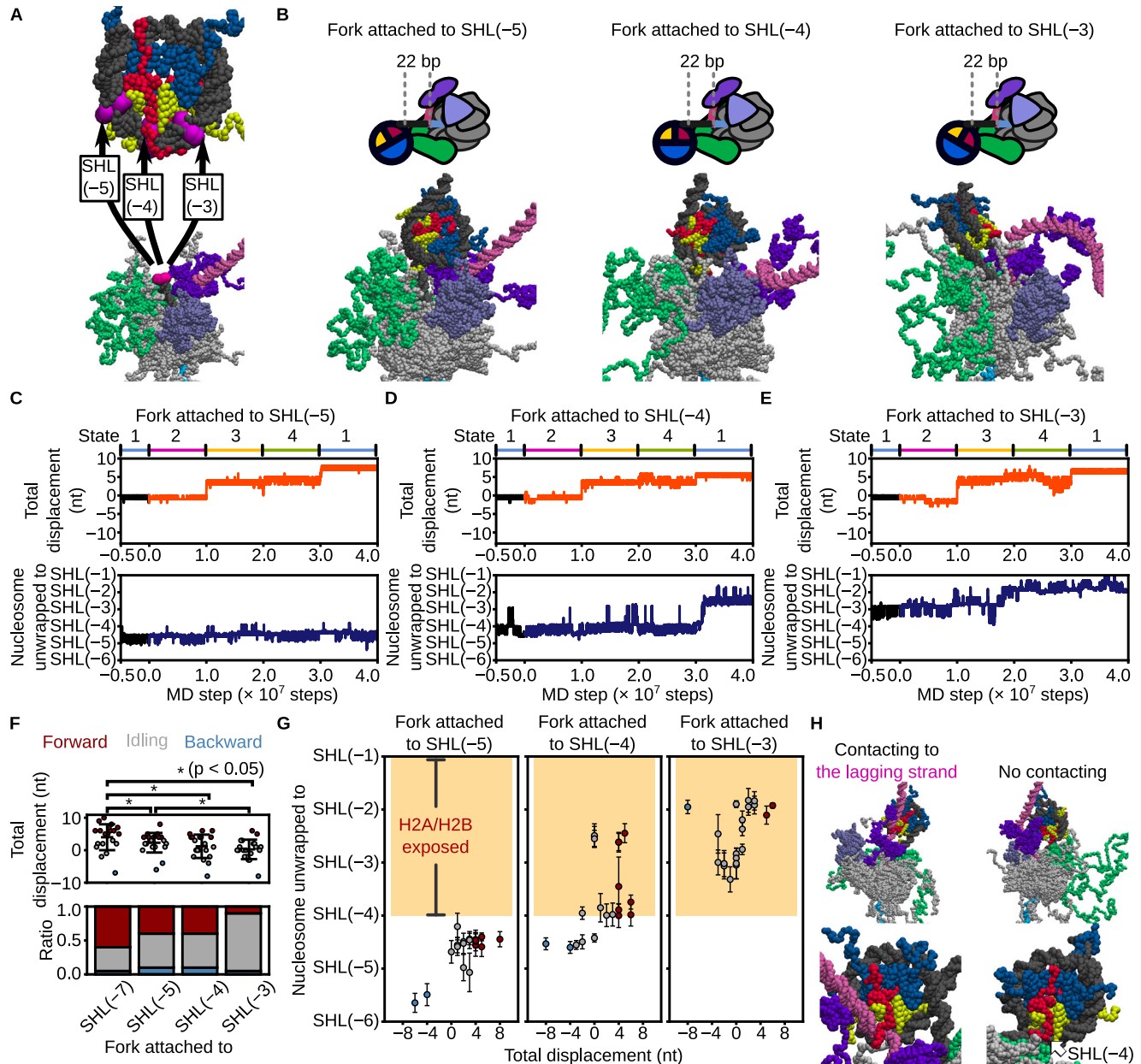

**Fig. 6 | CMG translocation initiated from different positions relative to the nucleosome reveals positional dependence of unwinding and unwrapping. A** Schematic showing superposition of Y-forked DNA onto nucleosomal DNA. Magenta spheres indicate nucleotide pairs used for structural alignment. **B** Initial structures for simulations with CMG positioned 22 bp upstream of SHL(−5), SHL(−4), or SHL(−3) (left to right). **C–E** Time trajectories of CMG displacement (top) and nucleosome unwrapping extent (bottom) for each initial position. The black trace indicates the trajectory from State1 used to initiate simulations. Orange and blue lines represent representative forward-translocating trajectories. **F** Final CMG displacements (top) and classification of trajectories (bottom): forward (> +3 bp, red), backward (<−3 bp, blue), and idling (−3 to +3 bp, gray). Asterisks indicate $P < 0.05$ by one-tailed Wilcoxon-Mann-Whitney test [$P = 0.03$ (Cohen's $d = 0.48$) between SHL(−7) and SHL(−5), $P = 0.007$ (Cohen's $d = 0.74$) between SHL(−7) and

SHL(−4), $P = 0.0007$ (Cohen's $d = 1.06$) between SHL(−7) and SHL(−3), and $P = 0.007$ (Cohen's $d = 0.68$) between SHL(−5) and SHL(−3)]. Error bars indicate mean ± standard deviation ($n = 20$ for all conditions). **G** Correlation between final displacement and unwrapping extent for simulations initiated from SHL(−5) (left), SHL(−4) (middle), and SHL(−3) (right). Red, gray, and blue circles represent forward, idling, and backward trajectories, respectively. Orange shaded area represents nucleosome unwrapping beyond SHL(−4), which allows the exposed H2A/H2B dimer to associate with the lagging strand. Error bars show mean ± standard deviation over the final 100 frames ($n = 100$). **H** Representative snapshot (top) and magnified view (bottom) showing association of the lagging strand with the exposed H2A/H2B dimer surface (left) and no association (right). Source data are provided as a Source Data file.

when FACT was present (Fig. 7F). These results suggest that FACT binding to partially unwrapped nucleosomes weakens DNA–histone interactions, thereby facilitating *S. cerevisiae* CMG translocation through chromatin.

As described above, in the absence of FACT, DNA unwrapping up to SHL(−4) or SHL(−3) exposed the H2A/H2B surface, enabling lagging-strand binding in 65% (13/20) and 20% (4/20) of trajectories,

respectively (Fig. 7G). In contrast, when FACT was present, no lagging-strand binding was observed in simulations starting from SHL(−4), and the frequency was reduced to 10% (2/20) at SHL(−3) (Fig. 7G). These results suggest that FACT prevents lagging-strand association with the exposed histone surface, likely by occupying the H2A/H2B interface via the IDR of Spt16, thereby suppressing direct histone transfer to the lagging strand.

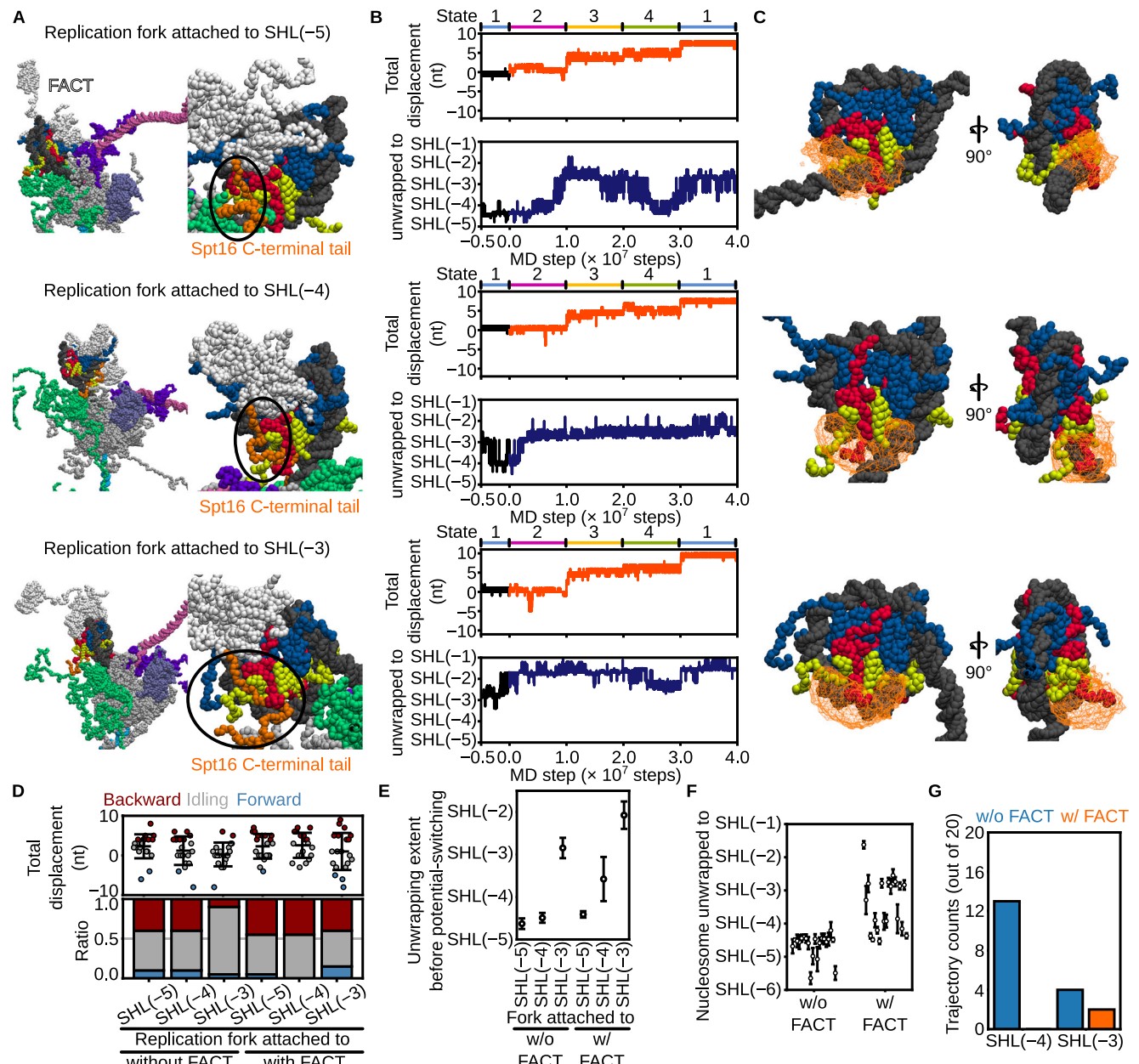

**Fig. 7 | FACT promotes CMG translocation through nucleosomes and suppresses lagging-strand histone recycling. A** Initial configurations of FACT-loaded nucleosomes aligned with the Y-forked DNA such that the replication fork is positioned 22 bp upstream of SHL(−5) (top), SHL(−4) (middle), or SHL(−3) (bottom). Orange beads indicate residues of the Spt16 C-terminal tail (residues 969–1035); white beads represent the remainder of FACT. **B** Time trajectories of CMG displacement (1st, 3rd, and 5th rows) and nucleosome unwrapping (2nd, 4th, and 6th rows) from simulations initiated at SHL(−5), SHL(−4), and SHL(−3), respectively. Black line shows the trajectory from State1 used to initiate simulations; orange and blue lines represent forward-translocating trajectories. **C** Isosurfaces of the spatial probability density of the Spt16 C-terminal tail (isovalue =

0.002 Å$^{-3}$), viewed from two orientations. **D** Final CMG displacements (top) and classification of trajectories (bottom) as forward (> +3 bp, red), backward (<−3 bp, blue), or idling (−3 to +3 bp, gray). Error bars indicate mean ± standard deviation (*n* = 20 for all conditions). **E** Nucleosome unwrapping extent at the beginning of simulations (State 1), averaged over 100 frames before potential switching. Error bars show mean ± standard deviation over the final 100 frames (*n* = 100). **F** Final nucleosome unwrapping extent in simulations initiated from SHL(−5), averaged over the last 100 frames. Error bars show mean ± standard deviation over the final 100 frames (*n* = 100). **G** Number of trajectories in which the lagging strand contacted the exposed surface of the H2A/H2B dimer, with and without FACT. Source data are provided as a Source data file.

## Discussion

Our simulations support the physicochemical plausibility of an asymmetric hand-over-hand mechanism for CMG helicase translocation. Using coarse-grained molecular dynamics simulations, we identified four distinct conformational states whose sequential transitions enable unidirectional 3′ → 5′ movement along ssDNA, consistent with biochemical observations that not all Mcm subunits require ATPase activity[12,17]. We further showed that RPA and Csm3/Tof1 enhance CMG

translocation via distinct mechanisms—clearing the lagging strand and stabilizing the parental duplex, respectively. In the context of chromatin, nucleosomes present a major obstacle to CMG progression, which is alleviated by FACT, while direct histone transfer to the lagging strand is suppressed. Together, these findings offer a mechanistic framework for understanding how helicase translocation, histone chaperone activity, and epigenetic inheritance are coordinated during eukaryotic DNA replication.

We identified four distinct DNA-binding conformations of CMG as the minimal set required for efficient $3' \rightarrow 5'$ translocation, with directional movement arising from sequential transitions among these states. While homo-hexameric helicases in bacteria[11], archaea[8], and viruses[9,10] typically employ a symmetric ATPase cycle tightly coupled to ssDNA binding, cryo-EM structures of *S. cerevisiae* CMG and *D. melanogaster* CMG have captured multiple conformational states with different ATP/ADP binding patterns, but these snapshots do not collectively resolve a symmetric, sequential rotation analogous to that seen in homo-hexameric systems[12,16,31]. In our simulations, we modeled state transitions by altering the energy landscape between four stable conformations, reflecting ATP-hydrolysis-driven conformational changes. Although we cannot exclude the possibility that ATP hydrolysis occurs in a sequential and symmetric manner with as-yet unresolved intermediates, our findings show that asymmetric conformational dynamics alone are sufficient to drive translocation. This supports an alternative, asymmetric mechanism consistent with biochemical observations that not all Mcm2–7 ATPase sites are essential for activity[12,17].

Our simulations revealed that RPA and Csm3/Tof1 enhance CMG-mediated dsDNA unwinding through distinct mechanisms, with a synergistic effect observed on poly-GC sequences but not on poly-AT sequences. This suggests that RPA alone is sufficient to promote unwinding of thermolabile DNA, while Csm3/Tof1 becomes critical under thermostable conditions. This is consistent with prior in vitro reconstitution studies showing that RPA is essential for robust DNA replication, whereas loss of Csm3/Tof1 has a more moderate impact[18,20,33]. These observations imply that the contributions of RPA and Csm3/Tof1 to replication may depend on the underlying DNA sequence context. In contrast, Csm3/Tof1, which directly interacts with CMG, may modulate its intrinsic helicase activity in a manner that is independent of DNA sequence. Further investigation will be important to fully elucidate the mechanistic role of Csm3/Tof1 in regulating CMG activity during replication.

Recent studies have shown that Csm3/Tof1 and FACT cooperatively facilitate replication through nucleosome arrays[34,43,44]. Our simulations revealed that Csm3/Tof1 promotes nucleosome unwrapping from the entry end up to SHL(−4.5). Meanwhile, previous studies have demonstrated that FACT preferentially binds to nucleosomes that are partially unwrapped at SHL(−4.5)[39]. Considering these findings, a plausible mechanism emerges in which Csm3/Tof1 induces partial unwrapping up to SHL(−4.5), creating a binding site for FACT at the exposed histone core complex. This suggests a molecular model for the cooperative functional interplay between Csm3/Tof1 and FACT. Importantly, this mechanism does not exclude the previously proposed cooperative model in which Tof1 and FACT interact directly[34,37]. Instead, our findings introduce a complementary mechanism that further reinforces the direct interaction model between Tof1 and FACT.

During CMG invasion into a nucleosome, we observed the association of the lagging strand with an H2A/H2B dimer exposed to the solvent. Previous studies have shown that during eukaryotic transcription through a nucleosome, upstream DNA forms a loop and interacts with the exposed H2A/H2B dimer, supporting direct histone recycling either upstream or to its original position[45,46]. Our simulation results suggest that parental histones could be directly recycled to the lagging strand during chromatin replication, similar to the process observed in transcription. However, previous our work has demonstrated that the histone chaperone Nap1 dismantles the exposed H2A/H2B dimer from a partially unwrapped nucleosome[25]. Additionally, in simulations with FACT, the Spt16 C-terminal tail prevented lagging-strand association with the H2A/H2B dimer. These findings suggest that histone chaperones may suppress the direct histone recycling pathway. Future studies involving direct visualization of DNA replication through a nucleosome will be essential to validate the direct recycling mechanism and its regulation by histone chaperones.

Our study provides experimentally testable predictions regarding the mechanism of *S. cerevisiae* CMG translocation and the roles of replication-associated factors. The simulations support an asymmetric hand-over-hand model, characterized by heterogeneous $3' \rightarrow 5'$ step sizes. High-resolution dual-trap optical tweezers, which have previously resolved stepping behavior in non-hexameric helicases such as UvrD[47], could be employed to validate heterogeneous stepping in *S. cerevisiae* CMG. On the other hand, in a symmetric hand-over-hand model, each MCM subunit engages two nucleotides of downstream DNA, leading to uniform step sizes. This would result in homogeneous translocation steps, which contrasts with the heterogeneous step sizes predicted by our asymmetric model and could be experimentally distinguished. Such assays could also directly test whether Csm3/Tof1 suppresses backtracking by quantifying the frequency of forward versus backward steps in its presence or absence. Furthermore, reconstitution of nucleosome-containing DNA substrates in single-molecule setups would enable investigation of FACT's role in assisting CMG progression. Monitoring nucleosome unwrapping and helicase advancement in the presence or absence of FACT would clarify whether FACT actively reduces the energy barrier, as predicted by our simulations. These experimental approaches would provide direct validation of our computational results and refine the mechanistic framework for chromatin replication.

While our simulations provide mechanistic insights into CMG translocation, DNA unwinding, and nucleosome traversal, several limitations warrant consideration. First, the coarse-grained model simplifies ATP hydrolysis dynamics and protein–DNA interactions, which may obscure transient intermediates and prevent direct mapping to physical timescales. Although an approximate time step (-10 fs) can be assigned, the smoothing of the energy landscape in coarse-grained models accelerates conformational transitions in a non-linear fashion. Furthermore, dwell times between CMG conformational states remain experimentally undetermined, making it difficult to extract absolute translocation velocities from our trajectories. Second, the coarse-grained model simplifies ATP hydrolysis dynamics and protein–DNA interactions, potentially overlooking transient intermediates. Second, essential replisome components—such as DNA polymerases and chromatin remodelers—were not included, although they likely influence CMG function in vivo. Third, although our results suggest that FACT promotes nucleosome unwrapping and modulates histone recycling, experimental validation is needed to confirm these roles in the context of a complete replisome. Despite these limitations, our findings offer a robust framework for understanding the coordination of helicase activity, nucleosome navigation, histone chaperone function, and histone recycling during eukaryotic DNA replication.

## Methods

### Coarse-grained molecular models

We used a coarse-grained protein model in which each amino acid is represented by a single bead located at the $C_\alpha$ atom. Inter-residue interactions were modeled using the AICG2+ potential, which stabilizes a given reference structure and captures native fluctuations[29]. For DNA, each nucleotide was modeled using three beads representing the base, sugar, and phosphate groups. The 3SPN.2 C potential was applied to preserve B-form geometry, reproduce sequence-dependent curvature, and accurately model the persistence lengths and melting temperatures of both ssDNA and dsDNA[30]. This protein–DNA modeling framework has been validated in previous studies[25,38,48–50] for its ability to reproduce key structural and interaction features.

Reference structures for *S. cerevisiae* CMG were constructed from three cryo-EM structures (PDB IDs: 8KG6, 8KG9, and 8W7M)[16], corresponding to DNA-binding States1, 2, and 3, respectively. State4 was generated via homology modeling using MODELLER[51], with *D.*

*melanogaster* CMG (PDB ID: 6RAY)[12] serving as the template. Reference models for Tof1, Csm3, and Ctf4 were obtained from the 8KG6 cryo-EM structure[16]. The α-helical region of Mrc1, which interacts with both CMG and Tof1, was modeled by homology using the cryo-EM structure of the replisome (PDB ID: 8B9A)[52]. IDRs, unresolved in the cryo-EM maps, were modeled using MODELLER[51] with random coil initial conformations.

The structural model of RPA used in this study was adopted from our previous work[26]. Yeast RPA is a heterotrimer composed of Rfa1, Rfa2, and Rfa3. The reference structure of Rfa1 was assembled from domain structures of DBD-F (PDB ID: 5OMB)[53], DBD-A (PDB ID: 1YNX)[54], DBD-B (homology model based on PDB ID: 1JMC)[55], and DBD-C (PDB ID: 6I52)[56]. For Rfa2, the winged-helix domain (homology model based on PDB ID: 4OUO)[57] and DBD-D (PDB ID: 6I52)[56] were used. Rfa3 was modeled based on the DBD-E structure (PDB ID: 6I52)[56]. The complete RPA model was constructed by linking individual domains with flexible linkers, following the approach established in our previous study[26].

The coarse-grained model of the histone octamer was constructed based on the X-ray crystal structure (PDB ID: 1KX5)[58]. Model parameters were adopted from previous studies calibrated to reproduce nucleosome unwrapping dynamics[38] and nucleosome disassembly[25,48].

FACT, a heterodimeric histone chaperone composed of Spt16 and Pob3, was modeled using a combination of crystal and cryo-EM structures. The N-terminal domain of Spt16 (residues 1–447) was based on the X-ray crystal structure (PDB ID: 3BIQ)[59], while the middle and C-terminal domains of both Spt16 and Pob3 were taken from the cryo-EM structure (PDB ID: 7NKY)[39]. The full-length FACT model was generated by linking these domains with flexible linker regions using MODELLER[51].

The 495-nucleotide ssDNA template was generated using 3DNA[60], with the *S. cerevisiae* CMG complex positioned 157 nt from the 5′ end. For simulations of fork progression, we constructed a Y-shaped DNA substrate comprising 168 bp (parental), 113 bp (lagging), and 285 bp (leading) of dsDNA, along with 29 nt and 42 nt of ssDNA on the lagging and leading strands, respectively. The ps1β loop of Mcm3, which engages the leading-strand ssDNA, was positioned 7 nt away from the fork junction in the initial configuration.

Intermolecular interactions were modeled primarily through excluded volume and electrostatic potentials. However, these were insufficient to stabilize several critical protein–DNA and protein–protein interfaces, including those between the histone octamer and dsDNA, FACT and dsDNA, FACT and the histone octamer, RPA and ssDNA, and Mcm2–7 and ssDNA. To address this, we incorporated additional structure-based potentials. For histone octamer–dsDNA interactions, coarse-grained hydrogen-bonding potentials[38] were applied to amino acid–phosphate pairs forming contacts in the reference structures (PDB IDs: 1KX5[58] and 3LZ0[61]). The interaction strengths were calibrated in previous studies to reproduce known binding affinities, nucleosome unwrapping profiles, and ionic strength dependence[38,48,62]. For FACT–dsDNA interactions, similar coarse-grained hydrogen-bonding potentials[38] were used. For FACT–histone octamer interactions, AICG2+ potentials[29] were applied between the octamer and the middle domains of Spt16 (residues 644–930) and Pob3 (residues 226–474), based on bead pairs extracted from the reference structure (PDB ID: 7NKY)[39]. For RPA–ssDNA interactions, we applied AICG2+ potentials[29] to residue–nucleotide pairs identified in the reference structure (PDB ID: 6I52)[56]. Experimental studies have shown that RPA binds to ssDNA with lifetimes on the order of tens of seconds to several minutes[63,64], which is significantly longer than the estimated time for the replisome to translocate by 12 nucleotides (~0.5 s)[18]. To ensure that RPA remains stably bound to ssDNA over the timescale of our simulations, we scaled the interaction strength between RPA and ssDNA. The resulting behavior is consistent

with experimental binding lifetimes and was validated in our previous work[26]. For Mcm2–7–ssDNA interactions, coarse-grained hydrogen-bonding potentials were derived from multiple reference structures (PDB IDs: 8KG6, 8KG9, 8W7M[16], and 6RAY[12]), with interaction strengths tuned to maintain the native DNA-binding configuration.

Electrostatic interactions were modeled using Debye–Hückel theory. Partial charges on protein surface beads were assigned using the RESPAC algorithm[65], which reproduces electrostatic surface potentials with near-atomistic accuracy, and has been successfully applied to simulate the dynamics of protein–DNA complexes[50,66]. These implementations help ensure that the absence of atomic detail does not compromise the mechanistic validity of our simulation results. For IDRs, where RESPAC cannot be applied, charges were assigned manually: +1e for lysine and arginine, and −1e for aspartic acid and glutamic acid. For DNA, phosphate groups were assigned a charge of −0.6e for intra-DNA interactions to account for counterion condensation along the backbone. For protein–DNA interactions, however, phosphate charges were set to −1.0e to reflect counterion release upon complex formation. The Debye–Hückel approximation is a simplified treatment, but it has been shown to be effective when combined with coarse-grained hydrogen-bonding potentials in reproducing salt-dependent nucleosome dynamics, including energetic barriers and unwrapping profiles[38,48]. Thus, it provides a sufficiently accurate representation of the energy landscape for modeling mesoscopic protein–DNA interactions in our CG-MD framework.

## Molecular dynamics simulations with potential switching

Conformational changes of *S. cerevisiae* CMG associated with ATP hydrolysis and ADP–ATP exchange were modeled by sequentially switching the underlying energy potentials[28]. Simulations were initiated in State1, and the AICG2+ potentials, hydrogen-bonding interactions, and RESPAC-derived surface charges specific to each state were sequentially updated to represent transitions through States2, 3, (4), and back to State 1.

Langevin dynamics with implicit solvent treatment was used to integrate the equations of motion with a timestep of 0.2 CafeMol time units (~0.01 ns), speeding up the simulations. Simulations were performed at a monovalent salt concentration of 300 mM, a temperature of 300 K, and a friction coefficient of 0.843. Although 150 mM salt is typically considered physiological, we used 300 mM monovalent salt to compensate for the limitations of the Debye–Hückel model, which does not explicitly account for divalent ion effects or local ion condensation near DNA. The dielectric constant of the solvent was set to ε = 72.2, computed using an empirical formula derived by Stogryn[67] as a function of temperature and ionic strength. Simulations without nucleosomes were carried out using CafeMol 3.2[68] [https://www.cafemol.org], whereas simulations including nucleosomes were performed using a developer version of OpenCafeMol[69]. The conformational switching interval for the CMG states was set to $1 \times 10^7$ steps. All the structural snapshots and movies of the simulations were visualized by PyMol[70] and VMD[71].

## Modeling of the *S. cerevisiae* CMG-FPC in complex with a nucleosome

Direct superposition of the leading edge of the *S. cerevisiae* CMG–FPC complex onto SHL(−7), SHL(−5), SHL(−4), or SHL(−3) of a nucleosome often resulted in steric clashes, making such configurations unsuitable as initial structures for simulations. To address this, we first performed coarse-grained molecular dynamics simulations of the nucleosome alone to sample a conformational ensemble. From this ensemble, we manually selected nucleosome conformations that avoided steric interference upon superposition with the *S. cerevisiae* CMG–FPC complex. The resulting structures were then subjected to energy minimization and further relaxed by a short simulation of $1 \times 10^7$ steps.

**Reporting summary**

Further information on research design is available in the Nature Portfolio Reporting Summary linked to this article.

## Data availability

The data that support the findings of this study are available within the article and Supplementary Information files. The input and trajectory files have been submitted to the Biological Structure Model Archive (BSM-Arc) under BSM-ID BSM00082 [https://doi.org/10.51093/bsm-00082]. Source data are provided with this paper.

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

## Acknowledgements

We would like to thank members of the Theoretical Biophysics laboratory at Kyoto University for discussions and assistance throughout this work. This work was supported by the Grant-in-Aid for Transformative Research Areas (24H00882 to T.T.), the grant from the Takeda Science Foundation (to T.T.), the grant from the Shimazu Science Foundation (to T.T.), the grant from the Inamori Foundation (to T.T.), the grant from the Yamada Science Foundation (to T.T.), the Grant-in-Aid for Japan Society for the Promotion of Science Fellows (22J21003 to F.N.) and the Grant-in-Aid for Scientific Research (B) (21H02441 to S.T.).

## Author contributions

F.N. and T.T. designed the project; F.N. performed the molecular dynamics simulations and analysis, Y.M., M.Y., and S.T. developed the OpenCafeMol software for the simulations, and F.N., Y.M., M.Y., S.T., and T.T. co-wrote the manuscript.

## Competing interests

The authors declare no conflict of interest.
