## [Transparent Peer Review file · Nature Communications]

Mechanistic models of asymmetric hand-over-hand translocation and nucleosome navigation by CMG helicase

Corresponding Author: Professor Tsuyoshi Terakawa

Version 0:

Reviewer comments:

Reviewer #2

(Remarks to the Author)

Review of manuscript # NCOMMS-25-33913

The manuscript details the results from a molecular modeling study that used coarse-grained molecular dynamics (CG-MD) to simulate translocation of the CMG helicase on single-stranded DNA, its ability to unwind DNA at a single-strand/double-strand junction and to navigate nucleosomes. The simulations were set up to alternate among four distinct DNA binding conformers of the CMG, which were experimentally observed by cryo-EM. This switching behavior of the simulation protocol was designed to mimic nucleotide-induced transitions of the helicase during its translocation cycle.

Understanding the translocation and unwinding mechanism of the CMG helicase is undoubtedly an important problem for the field of DNA replication. The simulation protocols use a coarse-grained methodology previously advanced and validated by the corresponding author's group. The use of CG-MD and the CafeMol program is justified by the timescales involved in ATP hydrolysis and ssDNA translocation by the CMG. From a technical standpoint the simulations appear to have been carried out with appropriate protocols and the description of the methodology is sufficiently detailed. However, I have a few reservations that prevent me from recommending the article in its present form. The primary concern is that the outcome of the simulations may be predicated on the initial selection of cryo-EM states. Omitting one or more relevant intermediate states could potentially change the simulation outcome by selectively increasing barriers for certain state-to-state transitions, thus affecting the likelihood of observing an asymmetric translocation mechanism. Second, the paper does not unambiguously discriminate between the asymmetric hand-over-hand mechanism and the symmetric rotary mechanism proposed for homo-hexameric helicases: "Our simulations support the physicochemical plausibility of an asymmetric hand-over-hand mechanism for CMG helicase translocation." Plausibility is not the same as producing a definite mechanism. Third, and perhaps less significant point is that there is no substantive attempt by the authors to analyze the structural aspects of the observed mechanisms. Thus, the structural elements responsible for translocation, nucleosomal unwrapping or interactions with various factors – RPA, Mrc1, Tof1, Csm3, Ctf4 – are not clearly defined. Indeed, all main figures show a representative snapshot followed by panels describing statistics of the observed displacement along the DNA or the forward, backward or idling behavior of CMG.

Therefore, I would ask the authors to consider the following points in revising the manuscript.

1) On page 3: "Among the six available configurations (previously labeled as I–VI) 16, we selected three representative conformations ... that exhibit distinct ssDNA-binding patterns (Fig. 1A). We excluded State VI, which lacks ssDNA engagement, State II due to its high similarity to State I, and State IV owing to the low resolution of its density map (7.3 Å), which could compromise accurate modeling of protein-DNA interactions."

Are the authors confident that they have included (or not selectively excluded) all relevant intermediate states? For instance, two states may not differ substantively in terms of DNA engagement and yet have important differences in other regions of the structure. Also, how do the authors exclude the possibility that the mechanism may involve a transient disengagement from DNA, in which case excluded State IV may be perfectly relevant.

2) One significant limitation of using a coarse-grain description for the protein–nucleic acid complex is that all explicit protein-DNA interactions, e.g., salt-bridge interactions to the DNA backbone are replaced with much less accurate bead-

bead interactions. What evidence do the authors present that the approach does not lead to artifacts.

3) "Protein–DNA interactions were modeled as hydrogen bonds that switched synchronously with the structure-stabilizing potential, while electrostatic and excluded-volume interactions remained unchanged throughout."
How does one represent H-bonding in this case? With no atomistic description only a distance cutoff can be enforced but not the precise H-bond geometry (angle between hydrogen and heavy atoms).

4) "We performed simulations of ScCMG bound to ssDNA using the conformational switching scheme, conducting 20 independent runs with different random seeds. In 70% of the trajectories (14/20), CMG translocated 12 ± 1 nucleotides in the 3'→5' direction after a single switching cycle (Fig. 1D, Supplementary Movie 1), far exceeding the displacement expected from passive diffusion without switching (Supplementary Fig. 1A)."
This may not be a completely fair comparison. By actively switching conformations for the ATPase ring in the middle of the CG-MD run energy is added artificially to the system. This energy may simply dissipate through the system and accelerate diffusion compared to the case where no such disruption is introduced. This may not necessarily imply directional motion as required for a molecular motor.

5) "Although these findings do not exclude the possibility of a symmetric hand-over-hand mechanism, previous biochemical evidence showing that not all Mcm2–7ATPase subunits are essential for translocation reinforces the plausibility of an asymmetric handover-hand model."
How would one test the symmetric hand over hand mechanistic hypothesis based on CG-MD modeling?

6) How do you know that partial DNA unwrapping from the nucleosome is not an artifact of the CG representation or the lack of accurate protein-DNA interactions in the CG force field?

7) Is Debye-Huckel treatment of electrostatics sufficient to accurately reproduce the energy landscape of protein-DNA complex formation? How do the author's conclusions depend on ionic strength?

8) A follow up question. Why did the authors use 300 mM salt concentration instead of 150 mM, which is considered physiological?

9) "For RPA-ssDNA interactions, we applied AICG2+ potentials to residue-nucleotide pairs observed in the reference structure (PDB ID: 6I52), with strengths adjusted to stabilize the complex. Adjusted how? This seems like an ad hoc scaling of interactions to prevent the complex from coming apart. Was there any validation done to justify the change in RPA parameters?"

10) On page 15: "Simulations were performed at a monovalent salt concentration of 300 mM, a temperature of 300K, and a friction coefficient of 0.843. The dielectric constant was computed based on ionic strength and temperature. What was the dielectric constant used? And what is the rationale for using it as an adjustable parameter?"

11) Is the switching between different CMG conformational states abrupt? To what extent does the switching disturb the sampling of the system?

Reviewer #3

(Remarks to the Author)

Nagae et al. used coarse-grained molecular dynamics simulations to investigate the translocation mechanism of the eukaryotic replicative helicase, CMG complex. They began with three distinct EM-derived yeast CMG-ssDNA conformations showing that insertion of an extra intermediate enhances ssDNA tracking. They then simulate dsDNA unwinding by CMG, revealing that backward slipping during the ATPase cycle is frequent, that unwinding on a random-sequence fork yields many unproductive translocation events, and that GC-rich sequences exacerbate idling while reducing net forward steps. They found inclusion of RPA prevent the ssDNA/dsDNA junction from entering and clogging the helicase pore, thereby boosting unwinding efficiency. Addition of the fork protection complex (Mrc1 and Csm3–Tof1) clamps the parental duplex ahead of CMG to suppress backward movement and improve net fork progression. In nucleosome-encounter simulations, CMG advances by peeling DNA partially off the histone core, a process facilitated by electrostatic interactions between Csm3–Tof1 and the nucleosome. Finally, inclusion of FACT in the model blocks the displaced lagging-strand DNA from binding exposed histone surfaces, reducing direct histone transfer onto the lagging strand.

This is a well-written manuscript with interesting findings. Before acceptance, however, I believe the following points should be addressed:

1. Although the authors focus on three of the six EM-determined CMG–ssDNA structures, it remains possible that CMG translocation proceeds via a symmetric hand-over-hand cycle involving all six states. They should expand their simulations to include each published conformation and discuss whether additional intermediates alter translocation pathways or efficiency.

2. The current metrics of "net translocation" and "step size" do not directly map onto experimentally measured unwinding rates. The authors should extract effective translocation velocities (e.g. bp s⁻¹) from their trajectories and compare them to reported ssDNA translocation rates (Wasserman et al., PMID: 31348887), fork unwinding without RPA (Burnham et al.,

PMID: 31089141), and with RPA (Kose et al., PMID: 32709841).

3. Previous in vitro work showed that binding of RPA to the excluded strand near the fork junction stimulates CMG unwinding by the same proposed mechanism (Kose et al., PMID: 32709841). The authors could also test this by replacing the 29-nt ssDNA arm in Figure 3 with duplex DNA and evaluating whether that likewise prevents central-channel clogging and enhances unwinding as seen in the in vitro unwinding assays by Kose et al.

4. Backward slipping of CMG in the absence of RPA was attributed to DNA re-annealing ahead of the helicase (Burnham et al., PMID: 31089141). Given the pronounced GC-dependent backtracking the authors observe (Fig. 3D), they should comment on whether strand re-hybridization contributes to this effect and, if possible, quantify base-pair reformation events in their simulations.

5. While the fork protection complex is known to accelerate replication forks, direct biochemical evidence for FPC stimulating CMG's intrinsic helicase activity is limited. The authors should acknowledge this gap and use their modeling pipeline to test whether clamping the duplex ahead of CMG might paradoxically stabilize backward slips if strand re-annealing drives reverse motion.

Version 1:

Reviewer comments:

Reviewer #2

(Remarks to the Author)

The authors have constructively addressed my previous comments. Potential caveats associated with the coarse-grained molecular dynamics (CG-MD) approach are more completely explained in the revised manuscript. The authors have also clarified the possible limitations arising from a limited number of intermediate states for the CMG helicase resolved by cryo-EM. With these additions to the manuscript, I recommend the paper for publication without further changes.

Reviewer #3

(Remarks to the Author)

Following revision, the authors ran additional simulations replacing the excluded-strand ssDNA with duplex DNA, which reproduced the experimentally observed stimulation mechanism. The remaining requests (comprehensive state coverage, velocity benchmarking, GC-dependent backtracking, and FPC effects) cannot be addressed conclusively given current structural availability and the inherent limits of coarse-grained MD. The authors have acknowledged these constraints and added clarifying text. In light of these reasonable constraints, I consider the revisions sufficient and recommend publication.

Open Access This Peer Review File is licensed under a Creative Commons Attribution 4.0 International License, which permits use, sharing, adaptation, distribution and reproduction in any medium or format, as long as you give appropriate credit to the original author(s) and the source, provide a link to the Creative Commons license, and indicate if changes were

made.

Responses to Reviewer #2

Comment 1:

The manuscript details the results from a molecular modeling study that used coarse-grained molecular dynamics (CG-MD) to simulate translocation of the CMG helicase on single-stranded DNA, its ability to unwind DNA at a single-strand/double-strand junction and to navigate nucleosomes. The simulations were set up to alternate among four distinct DNA binding conformers of the CMG, which were experimentally observed by cryo-EM. This switching behavior of the simulation protocol was designed to mimic nucleotide-induced transitions of the helicase during its translocation cycle. Understanding the translocation and unwinding mechanism of the CMG helicase is undoubtedly an important problem for the field of DNA replication. The simulation protocols use a coarse-grained methodology previously advanced and validated by the corresponding author's group. The use of CG-MD and the CafeMol program is justified by the timescales involved in ATP hydrolysis and ssDNA translocation by the CMG. From a technical standpoint the simulations appear to have been carried out with appropriate protocols and the description of the methodology is sufficiently detailed.

However, I have a few reservations that prevent me from recommending the article in its present form.

The primary concern is that the outcome of the simulations may be predicated on the initial selection of cryo-EM states. Omitting one or more relevant intermediate states could potentially change the simulation outcome by selectively increasing barriers for certain state-to-state transitions, thus affecting the likelihood of observing an asymmetric translocation mechanism.

Second, the paper does not unambiguously discriminate between the asymmetric hand-over-hand mechanism and the symmetric rotary mechanism proposed for homo-hexameric helicases: "Our simulations support the physicochemical plausibility of an asymmetric hand-over-hand mechanism for CMG helicase translocation." Plausibility is not the same as producing a definite mechanism.

Third, and perhaps less significant point is that there is no substantive attempt by the authors to analyze the structural aspects of the observed mechanisms. Thus, the structural elements responsible for translocation, nucleosomal unwrapping or interactions with various factors – RPA, Mrc1, Tof1, Csm3, Ctf4 – are not clearly defined. Indeed, all main figures show a representative snapshot followed by panels describing statistics of the observed displacement along the DNA or the forward, backward or idling behavior of CMG.

Therefore, I would ask the authors to consider the following points in revising the manuscript.

Response 1:

We sincerely thank the reviewer for his/her thoughtful and constructive evaluation of our manuscript. We appreciate the recognition of the importance of understanding the CMG helicase mechanism and the appropriateness of our coarse-grained molecular dynamics (CG-MD) approach. Below, we summarize our responses to the reviewer's main concerns. Issues raised have also been addressed in the point-by-point responses that follow.

Regarding the first and second concern, we fully agree that it is inherently difficult to guarantee that all functionally relevant intermediate states have been identified and our simulations demonstrate only plausibility. This limitation is shared by all structural and simulation-based studies. Nevertheless, significant progress in the field has been made by identifying a minimal set of conformational states sufficient to drive essential functions. Following this rationale, we initially tested three DNA-engaged states but found limited success in directional movement. Incorporating a fourth intermediate (State4) significantly improved translocation, suggesting that these four states represent a functionally sufficient set. While additional states may further facilitate conformational transitions, we believe the four-state cycle provides a robust backbone for modeling CMG dynamics. This point is further discussed in our response to Comment 2.

Regarding the third concern, the reviewer suggested that more detailed structural analysis of the observed mechanisms would strengthen the manuscript. In our original analysis, we primarily focused on RPA and Csm3/Tof1, describing how steric exclusion by RPA facilitates the removal of the lagging strand from the CMG pore, and how electrostatic interactions involving Csm3/Tof1 enhance CMG translocation and promote nucleosome unwrapping. As the reviewer correctly pointed out, we had not described structural roles for other replisome components, such as Mrc1 and Ctf4, in these processes. In response, we have now quantified the frequency of interactions between Mrc1/Ctf4 and DNA, as well as between Mrc1, Csm3/Tof1, Ctf4, RPA, and histones, as shown in Supplementary Figs. 5–7. To clarify these points, we have added the following sentences to the Results section:

“During translocation, Csm3/Tof1 directly engaged the parental dsDNA, consistent with previous cryo-EM observations²⁰, whereas Mrc1 and Ctf4 exhibited minimal interaction with DNA (Fig. 4J, K, and Supplementary Fig. 5).” and “Notably, during the ScCMG translocation, Csm3/Tof1 engaged the parental dsDNA in a manner similar to that shown in Fig. 4K, whereas interactions between Mrc1/Ctf4 and DNA, as well as between Mrc1, Csm3/Tof1, Ctf4, RPA, and histones, were rarely observed (Supplementary Fig. 6 & 7).”

Comment 2:

On page 3: “Among the six available configurations (previously labeled as I–VI) ¹⁶, we selected three representative conformations ... that exhibit distinct ssDNA-binding patterns (Fig. 1A). We excluded State VI, which lacks ssDNA engagement, State II due to its high similarity to State I, and State IV owing to the low resolution of its density map (7.3 Å), which could compromise accurate modeling of protein-DNA interactions.”

Are the authors confident that they have included (or not selectively excluded) all relevant intermediate states? For instance, two states may not differ substantively in terms of DNA engagement and yet have important differences in other regions of the structure. Also, how do the authors exclude the possibility that the mechanism may involve a transient disengagement from DNA, in which case excluded State IV may be perfectly relevant.

Response 2:

We thank the reviewer for raising this important question regarding the selection of cryo-EM conformational states. We fully agree that it is inherently difficult to guarantee that all functionally relevant intermediate states have been identified, especially given the transient nature of protein conformational dynamics. This limitation is shared by all structural and simulation-based studies, including those using cryo-EM or molecular dynamics (MD). Nonetheless, significant progress in the field has been made by identifying and testing the minimal set of conformational states sufficient to drive functions. In this study, we adopted a similar philosophy: rather than exhaustively sampling every possible intermediate, we selected, from the six cryo-EM structures currently available for yeast CMG, those that (i) exhibit distinct DNA engagement patterns, (ii) are of sufficiently high resolution to allow reliable modeling, and (iii) likely to contribute functionally to translocation.

Importantly, our simulation results show that using only three states (State1–3) led to a relatively low success rate of directional translocation, consistent with the reviewer’s concern that omitting intermediate states increases energetic barriers. In response to this limitation, we incorporated an additional intermediate state (State4), modeled by homology to a *Drosophila* CMG cryo-EM conformation. The inclusion of State4 substantially increased the success rate of translocation and resolved a kinetic bottleneck. Based on this, we conclude that the four-state model captures the minimal set of conformations required for directional CMG translocation. Of course, this does not

preclude the possibility that additional conformers—either currently unavailable or to be resolved in the future—may further smoothen the energy landscape and facilitate more efficient transitions. Even in such a case, we believe that the transitions among the four key states modeled here would still represent a fundamental backbone of the mechanism.

We also note that our simulation framework can, in principle, accommodate disengagement of DNA from the protein. During each conformational transition, the hydrogen-bonding interactions between DNA and the outgoing state are disabled, while those of the incoming state are enabled. This design inherently allows transient DNA disengagement, particularly in transitions involving minimal overlap between DNA-binding subunits. Thus, if transient disengagement plays a mechanistic role—as the reviewer suggests—it is at least partially captured within our model.

To clarify these points, we modified and added the sentences, “To identify a minimal set of conformational states sufficient to drive translocation, we excluded State VI because it lacks ssDNA engagement, State II due to its high structural similarity to State I (RMSD = 0.5 Å), and State IV because its low resolution (7.3 Å) may compromise accurate modeling of protein–DNA interactions.” and “Furthermore, this suggests that as-yet unresolved conformational states may also contribute to stabilizing and smoothing the transitions between DNA-bound states. Therefore, our findings do not definitively exclude the possibility of a symmetric hand-over-hand mechanism. Nonetheless, previous biochemical evidence demonstrating that not all Mcm2–7 ATPase subunits are essential for translocation^{12,17} supports the plausibility of an asymmetric hand-over-hand model.” in the Result section.

Comment 3:

One significant limitation of using a coarse-grain description for the protein–nucleic acid complex is that all explicit protein–DNA interactions, e.g., salt-bridge interactions to the DNA backbone are replaced with much less accurate bead-bead interactions. What evidence do the authors present that the approach does not lead to artifacts.

Response 3:

We thank the reviewer for pointing out a critical aspect of coarse-grained (CG) modeling—namely, the potential loss of atomic-level specificity in protein–DNA interactions. We agree that CG

models trade atomic detail for enhanced sampling and longer timescale dynamics. However, we would like to emphasize two key points that support the validity of our simulation.

In our simulation framework, protein–DNA interactions are explicitly incorporated via hydrogen-bonding potentials or AICG2+ contact terms. These are defined based on contacts observed in cryo-EM or crystal structures (PDB IDs: 8KG6, 8KG9, 8W7M, 1KX5, 6I52, etc.), ensuring that the interactions between DNA and amino acid residues are preserved in both geometry and specificity. The interaction strengths were further tuned in prior studies to reproduce binding affinities, DNA unwrapping profiles, and salt-dependence [e.g., Niina et al. (2017) PLoS Comput. Biol. 13:e1005880; Brandani et al. (2021) PLoS Comput. Biol. 17:e1009253; Yamauchi et al. (2025) eLife: <https://doi.org/10.7554/eLife.106752>]. Thus, although our model lacks explicit atoms, it retains a substantial degree of structure-specific interaction fidelity.

Furthermore, to compensate for the absence of atomic-level electrostatics, we employed RESPAC-derived partial charges on protein surfaces and modeled electrostatic interactions using the Debye–Hückel potential. While this is an approximation, prior studies [Terakawa & Takada (2014) J. Chem. Theory Comput. 10:711] have demonstrated that RESPAC can reproduce the electrostatic surface potential of DNA-binding proteins with atomistic details. Also, this treatment successfully reproduced the dynamics of protein-DNA complexes [Tan et al. (2016) J. Am. Chem. Soc. 138:8512; Inoue et al. (2022) Biophys. Physicobiol. 19:e190015].

To clarify these points, we modified and added the sentences, “**The interaction strengths were calibrated in previous studies to reproduce known binding affinities, nucleosome unwrapping profiles, and ionic strength dependence [Niina et al., 2017; Brandani et al., 2021; Yamauchi et al., 2025].**” and “**Electrostatic interactions were modeled using Debye–Hückel theory. Partial charges on protein surface beads were assigned using the RESPAC algorithm⁶⁴, which reproduce electrostatic surface potentials with near-atomistic accuracy, and has been successfully applied to simulate the dynamics of protein–DNA complexes [Tan et al., 2016; Inoue et al., 2022].**” in the Discussion section.

Comment 4:

“Protein–DNA interactions were modeled as hydrogen bonds that switched synchronously with the structure-stabilizing potential, while electrostatic and excluded-volume interactions remained unchanged throughout.”

How does one represent H-bonding in this case? With no atomistic description only a distance cutoff can be enforced but not the precise H-bond geometry (angle between hydrogen and heavy atoms).

Response 4:

We thank the reviewer for the opportunity to clarify this point. In our coarse-grained model, hydrogen-bonding interactions between proteins and DNA are represented by a structure-based potential that depends on one distance term and two angular terms. Specifically:

- The distance term is defined between the coarse-grained bead of an amino acid (donor) and the phosphate group of a DNA nucleotide (acceptor).
- The first angle term measures the orientation of the amino acid backbone and is defined by the angle formed between the vector connecting the amino acid to the phosphate and the vector between its neighboring amino acid beads ($i-1$, i , and $i+1$).
- The second angle term measures the geometry on the DNA side, defined by the angle between the vector from the amino acid to the phosphate and the vector connecting the phosphate to the sugar bead within the same nucleotide.

The equilibrium values for these distance and angle terms are taken directly from experimentally determined structures, ensuring that the native spatial arrangement of the protein–DNA interface is preserved. This representation enables us to encode hydrogen-bond-like interactions within a coarse-grained framework, approximating the geometry observed in atomistic structures. This method was originally validated previously [Niina et al. (2017) PLoS Comput. Biol. 13:e1005880].

In the revised manuscript, we now refer to this approach as **a coarse-grained hydrogen-bonding potential** to clearly distinguish it from conventional all-atom hydrogen bonds involving explicit hydrogens.

Comment 5:

“We performed simulations of ScCMG bound to ssDNA using the conformational switching scheme, conducting 20 independent runs with different random seeds. In 70% of the trajectories (14/20), CMG translocated 12 ± 1 nucleotides in the 3'→5' direction after a single switching cycle (Fig. 1D, Supplementary Movie 1), far exceeding the displacement expected from passive

diffusion without switching (Supplementary Fig. 1A).”

This may not be a completely fair comparison. By actively switching conformations for the ATPase ring in the middle of the CG-MD run energy is added artificially to the system. This energy may simply dissipate through the system and accelerate diffusion compared to the case where no such disruption is introduced. This may not necessarily imply directional motion as required for a molecular motor.

Response 5:

Reviewer 2 has raised an important point regarding the potential artifact introduced by abrupt conformational switching. We fully agree that switching the energy landscape between conformational states could, in principle, inject artificial energy into the system. To address this concern, we conducted a control set of simulations in which only the structural conformation of CMG was switched, while the hydrogen-bonding potentials responsible for DNA engagement were held fixed. In these control simulations, the final displacement of CMG after a complete switching cycle was 0 ± 6 nucleotides (new Supplementary Fig. 1E), in stark contrast to the directional 12 ± 1 nucleotide displacement observed when both structure and DNA-binding potentials were switched. This result demonstrates that the artificial energy introduced by conformational switching alone is not sufficient to drive unidirectional motion.

To clarify these points, we modified and added the sentences, “**Because the abrupt conformational switching introduces energy into the system, one might be concerned that the observed 12-nt translocation was driven primarily by this artificial energy input. To address this, we performed control simulations in which only the conformation of CMG was switched, while the coarse-grained hydrogen-bonding potentials responsible for DNA engagement were held fixed. Under these conditions, the final displacement of CMG after one complete cycle was 0 ± 6 nt (Supplementary Fig. 1E), indicating that the directional translocation observed in our original simulations is not attributable to the artificial energy introduced by conformational switching alone.**” in the Result section.

Comment 6:

“Although these findings do not exclude the possibility of a symmetric hand-over-hand mechanism, previous biochemical evidence showing that not all Mcm2–7 ATPase subunits are

essential for translocation reinforces the plausibility of an asymmetric handover-hand model.”

How would one test the symmetric hand over hand mechanistic hypothesis based on CG-MD modeling?

Response 6:

We appreciate the reviewer’s suggestion to further elaborate on how our simulation-derived hypotheses could be tested experimentally, particularly regarding the possibility of a symmetric hand-over-hand mechanism.

As discussed in the revised manuscript, one key prediction of our coarse-grained model is that the asymmetric hand-over-hand mechanism would produce heterogeneous step sizes during translocation, as different conformational transitions involve varying numbers of DNA-engaging subunits. This prediction could be experimentally tested using high-resolution dual-trap optical tweezers, which have previously resolved single-nucleotide steps in other helicases [e.g., UvrD; Carney et al. (2021) Nat Commun 12:7015].

In contrast, a symmetric hand-over-hand mechanism is expected to produce uniform stepping behavior, because each Mcm2–7 subunit binds two nucleotides and would contribute equally in a strictly rotational ATPase cycle. Therefore, if ScCMG were to exhibit homogenous stepping with uniform step sizes, this would support a symmetric model.

To clarify these points, we modified and added the sentences, “**On the other hand, in a symmetric hand-over-hand model, each MCM subunit engages two nucleotides of downstream DNA, leading to uniform step sizes. This would result in homogeneous translocation steps, which contrasts with the heterogeneous step sizes predicted by our asymmetric model and could be experimentally distinguished.**” in the Result section.

Comment 7:

How do you know that partial DNA unwrapping from the nucleosome is not an artifact of the CG representation or the lack of accurate protein-DNA interactions in the CG force field?

Response 7:

We thank the reviewer for raising this important concern. We agree that a potential limitation of coarse-grained modeling is the risk of introducing artifacts. However, we believe the observed partial unwrapping of nucleosomal DNA reflects mechanistically relevant behavior, for the following reasons.

First, the parameters of the coarse-grained hydrogen-bonding potentials used for nucleosome modeling were carefully calibrated in previous work to quantitatively reproduce the experimentally observed salt-dependence of nucleosome unwrapping and disassembly dynamics [Niina et al. (2017) PLoS Comput. Biol. 13:e1005880; Brandani et al. (2021) PLoS Comput. Biol. 17:e1009253]. These prior validations support the reliability of our model. Second, in the current simulations, the nucleosomal DNA remained stably wrapped up to superhelical location (SHL) -6.5 throughout the early stages of the trajectory. This indicates that the force field does not inherently destabilize the nucleosome. Third, the partial unwrapping observed in our simulations clearly correlated with the approach of CMG toward the nucleosome. This indicates that the unwrapping was driven by mechanical interactions with CMG.

To clarify these points, we modified and added the sentences, “At the early stage of the simulation (up to 1.0×10^7 steps; corresponding to ~ 1 nt of translocation), the nucleosome remained stably wrapped up to SHL(-6.5), although transient unwrapping and rewinding events were occasionally observed between SHL(-7) and SHL(-5.5) [Fig. 5C (i), (ii)], consistent with simulations conducted in the absence of ScCMG–FPC³⁸” in the Result section.

Comment 8:

Is Debye-Huckel treatment of electrostatics sufficient to accurately reproduce the energy landscape of protein-DNA complex formation? How do the author's conclusions depend on ionic strength?

Response 8:

We thank the reviewer for raising this important point. While we recognize that the Debye–Hückel (DH) approximation is a simplified treatment of electrostatics, it has been successfully applied in combination with coarse-grained hydrogen-bonding potentials in previous studies of the salt-dependent nucleosome unwrapping dynamics, including the location of the energetic barriers and the extent of unwrapping [Niina et al. (2017) PLoS Comput. Biol. 13:e1005880; Brandani et al. (2021) PLoS Comput. Biol. 17:e1009253]. Thus, it provides a sufficiently accurate approximation

of the energy landscape to model mesoscopic protein–DNA interactions within our CG-MD framework.

As the reviewer correctly points out, the efficiency of CMG invasion into the nucleosome is likely to depend on the ionic strength, since electrostatic screening modulates DNA–histone interactions. However, our simulations were performed under a monovalent salt concentration of 300 mM, which we consider to approximate physiologically relevant conditions. The rationale for this choice is provided in our response to Comment 9. While variations in ionic strength may modulate the efficiency of individual events, the qualitative insights obtained under the physiological conditions remain biologically meaningful.

To clarify these points, we modified and added the sentences, “**The Debye–Hückel approximation is a simplified treatment, but it has been shown to be effective when combined with coarse-grained hydrogen-bonding potentials in reproducing salt-dependent nucleosome dynamics, including energetic barriers and unwrapping profiles. Thus, it provides a sufficiently accurate representation of the energy landscape for modeling mesoscopic protein–DNA interactions in our CG-MD framework.**” in the Method section.

Comment 9:

A follow up question. Why did the authors use 300 mM salt concentration instead of 150 mM, which is considered physiological?

Response 9:

We thank the reviewer for this important follow-up question. While 150 mM is often cited as the physiological ionic strength, we intentionally used a higher monovalent salt concentration (300 mM) in our simulations to compensate for limitations of the Debye–Hückel (DH) electrostatics model.

The DH model treats electrostatic screening in mean-field manner and does not account for important features of multivalent ions—particularly the localization of divalent cations (e.g., Mg^{2+}) near the DNA phosphate backbone, which enhances local charge screening. Therefore, DH model using 150 mM monovalent salt would underestimate the effective screening strength. To partially correct for this, we used 300 mM monovalent salt as a first-order approximation to mimic the

stronger electrostatic screening provided by a mixed ion environment. This approach is consistent with previous coarse-grained simulation studies of nucleosome dynamics using the same modeling framework [Niina et al. (2017) PLoS Comput. Biol. 13:e1005880; Brandani et al. (2021) PLoS Comput. Biol. 17:e1009253].

To clarify these points, we modified and added the sentences, “Although 150 mM salt is typically considered physiological, we used 300 mM monovalent salt to compensate for the limitations of the Debye–Hückel model, which does not explicitly account for divalent ion effects or local ion condensation near DNA.” in the Method section.

Comment 10:

“For RPA-ssDNA interactions, we applied AICG2+ potentials to residue-nucleotide pairs observed in the reference structure (PDB ID: 6I52), with strengths adjusted to stabilize the complex. Adjusted how? This seems like an ad hoc scaling of interactions to prevent the complex from coming apart. Was there any validation done to justify the change in RPA parameters?”

Response 10:

We thank the reviewer for raising this important point. We agree that arbitrary adjustment of interaction strengths in coarse-grained models must be carefully justified.

In our simulations, the AICG2+ potentials for RPA–ssDNA interactions were derived from residue–nucleotide contacts observed in the reference structure (PDB ID: 6I52), following the same structure-based approach used throughout the model. However, because coarse-grained models lack explicit hydrogen atoms and detailed electrostatics, native contact-based potentials alone sometimes underestimate binding stability.

To guide the scaling of interaction strength, we referred to experimental measurements of RPA binding kinetics: Gibb et al. [(2012) Anal. Chem. 84:7607] showed that yeast RPA remains stably bound to ssDNA for over 60 minutes in single-molecule assays. Chen et al. [(2016) Nucleic Acids Res. 44:5758] reported residence times of ~50 seconds. In contrast, the time required for the replisome to progress by 12 nucleotides is estimated to be ~0.5 seconds in vitro [Yeeles et al., (2017) Molecular Cell 65:105].

Given this disparity in timescales, we reasoned that the RPA–ssDNA complex should remain intact during the course of our simulations. Therefore, we scaled the interaction strength such that the complex remained stably bound over the simulated timeframe, but without over-stabilizing it to the point of artificial rigidity. The same approach was successfully used in our previous work [Nagae et al. (2024) Nat Commun 15:9485], where simulated RPA–ssDNA binding behavior reproduced experimentally consistent patterns.

To clarify these points, we modified and added the sentences, “RPA stably binds to ssDNA for a minute-timescale^{62,63}, which is substantially longer than the time required for the replisome to translocate 12 nucleotides¹⁸ (~0.5 sec). To prevent unphysical dissociation during simulations, we scaled the interaction strength between RPA and ssDNA to maintain stable binding over the simulated timescale, while preserving conformational flexibility. The resulting behavior is consistent with experimental binding lifetimes and was validated in our previous work.” in the Result section.

Comment 11:

“Simulations were performed at a monovalent salt concentration of 300 mM, a temperature of 300K, and a friction coefficient of 0.843. The dielectric constant was computed based on ionic strength and temperature. What was the dielectric constant used? And what is the rationale for using it as an adjustable parameter?”

Response 11:

We thank the reviewer for this question. In our simulations, the dielectric constant of the solvent was set to $\epsilon = 72.2$, based on its dependence on both temperature and ionic strength. Rather than using a fixed dielectric constant (e.g., 78.0 for pure water at 298 K), we computed ϵ using the empirical formula derived by Stogryn [(1971) IEEE Trans. Microw. Theory Tech. 19:733], which accounts for the effects of temperature (T , in K) and ionic strength (C , in mol/L): $\epsilon(T, C) = (249.4 - 0.788 T + 7.20 \times 10^{-4} T^2) \times (1 - 0.2551 C + 5.151 \times 10^{-2} C^2 - 6.889 \times 10^{-3} C^3)$. At the simulation conditions of $T = 300$ K and $C = 0.3$ M monovalent salt, this equation yields $\epsilon \approx 72.2$, which we used throughout our simulations.

To clarify these points, we modified and added the sentences, “The dielectric constant of the solvent was set to $\epsilon = 72.2$, computed using an empirical formula derived by Stogryn⁶⁷ as a function of temperature and ionic strength.” in the Method section.

Comment 12:

Is the switching between different CMG conformational states abrupt? To what extent does the switching disturb the sampling of the system?

Response 12:

We thank the reviewer for this important question. In our simulations, the transitions between CMG conformational states were implemented as abrupt switches in the underlying potential energy function, as implemented in structure-based coarse-grained models we have developed for simulating conformational cycles [Koga et al. (2006) PNAS 103:5367].

To evaluate whether this abrupt switching perturbed the conformational sampling of the system, we computed the Q-score at each state transition (new Supplementary Figure 1A–C and 1G). The Q-score is defined as the ratio of the number of residue–residue contacts formed in the simulation to those in the reference structure, and serves as a metric for structural similarity to the target state.

For all transitions—State1→2, 2→3, 3→4, and 4→1—the Q-score rapidly recovered to stable values within a short timescale after switching, indicating that the system quickly relaxed into the corresponding energy basin and that the conformational ensemble was not significantly disrupted. This suggests that the effect of abrupt switching on conformational sampling is minimal.

To clarify these points, we modified and added the sentences, “At each state transition, the underlying potential energy function was abruptly switched, after which the ScCMG conformation quickly relaxed and reached equilibrium (Supplementary Fig. 1A–C).” and “At the transition to State4, the underlying potential energy function was abruptly switched, after which the ScCMG conformation quickly relaxed and reached equilibrium (Supplementary Fig. 1G).” in the Result section.

Responses to Reviewer #3

Comment 1:

Nagae et al. used coarse-grained molecular dynamics simulations to investigate the translocation mechanism of the eukaryotic replicative helicase, CMG complex. They began with three distinct EM-derived yeast CMG-ssDNA conformations showing that insertion of an extra intermediate enhances ssDNA tracking. They then simulate dsDNA unwinding by CMG, revealing that backward slipping during the ATPase cycle is frequent, that unwinding on a random-sequence fork yields many unproductive translocation events, and that GC-rich sequences exacerbate idling while reducing net forward steps. They found inclusion of RPA prevent the ssDNA/dsDNA junction from entering and clogging the helicase pore, thereby boosting unwinding efficiency. Addition of the fork protection complex (Mrc1 and Csm3–Tof1) clamps the parental duplex ahead of CMG to suppress backward movement and improve net fork progression. In nucleosome-encounter simulations, CMG advances by peeling DNA partially off the histone core, a process facilitated by electrostatic interactions between Csm3–Tof1 and the nucleosome. Finally, inclusion of FACT in the model blocks the displaced lagging-strand DNA from binding exposed histone surfaces, reducing direct histone transfer onto the lagging strand.

This is a well-written manuscript with interesting findings. Before acceptance, however, I believe the following points should be addressed.

Response 1:

We thank Reviewer 3 for the careful and thoughtful review of our manuscript, and for their interest in our work. In response to the comments and suggestions provided, we have thoroughly revised the manuscript. We believe that the revised version meets the standards for publication in Nature Communications.

Comment 2:

Although the authors focus on three of the six EM-determined CMG–ssDNA structures, it remains possible that CMG translocation proceeds via a symmetric hand-over-hand cycle involving all six states. They should expand their simulations to include each published conformation and discuss whether additional intermediates alter translocation pathways or efficiency.

Response 2:

We thank Reviewer 3 for this insightful comment. As we also discussed in our response to Reviewer 2, it is inherently difficult in structure- or simulation-based studies to guarantee that all functionally relevant intermediate states have been captured. Nonetheless, progress in the field has often been made by identifying a minimal subset of conformational states that are sufficient to reproduce biological functions. Following this rationale, we focused on a reduced set of cryo-EM structures of yeast CMG–ssDNA complexes. Among the six available structures, only four could be classified into functionally distinct DNA-binding states. Notably, all other published CMG–DNA conformations [Georgescu et al. (2017) Proc. Natl. Acad. Sci.; Goswami et al. (2018) Nat. Commun.; Eickhoff et al. (2019) Cell Reports; Baretić et al. (2020) Mol. Cell; Rzechorzek et al. (2020) Nucl. Acids Res.; Yuan et al. (2020) Nat. Commun.; Jones et al. (2021) EMBO J.; Jenkyn-Bedford et al. (2021) Nature; Jones et al. (2023) Mol. Cell; Xu et al. (2023) Nat. Commun.; Xia et al. (2023) Science; Henrikus et al. (2024) Nat. Struct. Mol. Biol.; Li et al. (2024) Nature; Batra et al. (2025) Science] reported to date resemble one of these four states.

A symmetric hand-over-hand mechanism would require six structurally distinct conformational states arranged with pseudo-sixfold symmetry, each engaging DNA in a rotationally ordered manner. However, to date, the available cryo-EM data do not support the existence of six such functionally distinct states. As such, we are currently unable to implement a symmetric model. Nevertheless, we fully agree that additional intermediate states could serve to smoothen transitions. In fact, we introduced a fourth intermediate state (State4) via homology modeling, which improved directional translocation.

To clarify these points, and as also described in our response to Reviewer 2, we modified and added the sentences, “To identify a minimal set of conformational states sufficient to drive translocation, we excluded State VI because it lacks ssDNA engagement, State II due to its high structural similarity to State I (RMSD = 0.5 Å), and State IV because its low resolution (7.3 Å) may compromise accurate modeling of protein–DNA interactions.” and “Furthermore, this suggests that as-yet unresolved conformational states may also contribute to stabilizing and smoothing the transitions between DNA-bound states. Therefore, our findings do not definitively exclude the possibility of a symmetric hand-over-hand mechanism. Nonetheless, previous biochemical evidence demonstrating that not all Mcm2–7 ATPase subunits are essential for translocation^{12,17} supports the plausibility of an asymmetric hand-over-hand model.” in the Result section.

Comment 3:

The current metrics of “net translocation” and “step size” do not directly map onto experimentally measured unwinding rates. The authors should extract effective translocation velocities (e.g. bp s^{-1}) from their trajectories and compare them to reported ssDNA translocation rates (Wasserman et al., PMID: 31348887), fork unwinding without RPA (Burnham et al., PMID: 31089141), and with RPA (Kose et al., PMID: 32709841).

Response 3:

We thank the reviewer for this thoughtful suggestion. We agree that comparing simulation-derived translocation velocities with experimentally measured unwinding rates would be informative. However, there are two major reasons why a direct mapping is currently not feasible.

First, although a rough timescale can be estimated by mapping one MD step to ~10 femtoseconds, the use of coarse-grained representations smooths the energy landscape and eliminates high-frequency fluctuations present in atomistic systems. This simplification accelerates conformational transitions in a way that is not linearly scalable to physical time. Second, the dwell times between conformational states of CMG have not been experimentally determined. Our simulation model implements transitions by switching between pre-defined potentials. As such, the time between transitions in our simulations cannot be interpreted as corresponding to real-world waiting times between ATPase-driven steps.

For these reasons, our simulations focus on identifying which conformational transitions are capable of producing translocation, and how accessory factors influence the efficiency of such motion. However, we agree that future work incorporating more realistic kinetic models may enable more accurate prediction of translocation velocities and direct comparison with studies [Wasserman et al. (2019) Cell 178:600, Burnham et al. (2019) Nat Commun 10:2159, Kose et al. (2020) Nat Commun 11:3713].

To clarify these points, we modified and added the sentences, “**First, the coarse-grained model simplifies ATP hydrolysis dynamics and protein–DNA interactions, which may obscure transient intermediates and prevents direct mapping to physical timescales. Although an approximate time step (~10 fs) can be assigned, the smoothing of the energy landscape in coarse-grained models accelerates conformational transitions in a non-linear fashion. Furthermore, dwell times between**

CMG conformational states remain experimentally undetermined, making it difficult to extract absolute translocation velocities from our trajectories.” in the Discussion section.

Comment 4:

Previous in vitro work showed that binding of RPA to the excluded strand near the fork junction stimulates CMG unwinding by the same proposed mechanism (Kose et al., PMID: 32709841). The authors could also test this by replacing the 29-nt ssDNA arm in Figure 3 with duplex DNA and evaluating whether that likewise prevents central-channel clogging and enhances unwinding as seen in the in vitro unwinding assays by Kose et al.

Response 4:

We thank Reviewer 3 for the insightful suggestion to further test whether central-channel clogging by the lagging strand inhibits CMG translocation. As suggested, we performed additional simulations in which the 29-nucleotide single-stranded DNA (ssDNA) region of the lagging strand was replaced with duplex DNA. We then analyzed the final displacement of CMG and the extent of lagging-strand intrusion into the central pore (new Supplementary Fig. 4).

Consistent with the proposed mechanism and with previous in vitro observations by Kose et al. [(2020) Nat. Commun. 11:3713], we found that duplex formation reduced the frequency of lagging-strand clogging and significantly enhanced CMG translocation. Specifically, simulations with the duplex lagging strand showed increased final displacements. In parallel, the minimum distance between the lagging strand and the CMG pore increased, indicating reduced intrusion.

To clarify these points, we modified and added the sentences, “To further test whether clogging of the central pore by lagging-strand ssDNA inhibits CMG translocation, we replaced the 29-nt single-stranded region of the lagging strand with duplex DNA and performed 20 simulations of unwinding poly-GC sequences (Supplementary Fig. 4C, 4D). Compared to the condition with exposed ssDNA, this duplex configuration resulted in significantly greater final displacements ($p < 0.05$, Wilcoxon–Mann–Whitney test; Supplementary Fig. 4E). Moreover, the lagging-strand DNA was more effectively excluded from the CMG pore when occluded by duplex formation than when exposed as ssDNA (Supplementary Fig. 4F). Together, these results indicate that preventing lagging-strand clogging—either through duplex formation or RPA binding—enhances CMG translocation, consistent with findings from a previous study¹⁹.” in the Result section.

We appreciate the reviewer's suggestion, which helped us further validate the proposed mechanism using an orthogonal test.

Comment 5:

Backward slipping of CMG in the absence of RPA was attributed to DNA re-annealing ahead of the helicase (Burnham et al., PMID: 31089141). Given the pronounced GC-dependent backtracking the authors observe (Fig. 3D), they should comment on whether strand re-hybridization contributes to this effect and, if possible, quantify base-pair reformation events in their simulations.

Response 5:

We thank Reviewer 3 for the valuable suggestion to examine whether reannealing of base pairs contributes to the backtracking. This is an important mechanism previously highlighted in experimental studies [Burnham et al. (2019) Nat Commun 10:2159].

In our simulations, we quantified base-pair reformation events at the dsDNA/ssDNA junction across different sequence contexts. As shown in new Supplementary Fig. 3A, reannealing events were more frequent in the poly-GC sequences than in the poly-AT sequences. This observation supports the notion that higher base-pairing stability promotes re-hybridization, which in turn can lead to central-pore clogging by the lagging strand and increase the likelihood of backtracking.

To clarify these points, we modified and added the sentences, “**Reannealing at the dsDNA/ssDNA junction was observed more frequently in simulations with the poly-GC sequence than with the poly-AT sequence (Supplementary Fig. 3A).**” in the Result section.

Comment 6:

While the fork protection complex is known to accelerate replication forks, direct biochemical evidence for FPC stimulating CMG's intrinsic helicase activity is limited. The authors should acknowledge this gap and use their modeling pipeline to test whether clamping the duplex ahead of CMG might paradoxically stabilize backward slips if strand re-annealing drives reverse motion.

Response 6:

We thank the reviewer for this thoughtful suggestion. We agree that the fork protection complex (FPC) may not only stabilize the parental dsDNA to suppress backtracking, but also potentially modulate the intrinsic helicase activity of CMG. However, in our current simulation framework, conformational transitions are implemented by switching between pre-defined potential energy functions rather than by explicitly modeling ATP hydrolysis events. As a result, our model does not capture how chemical energy from ATP hydrolysis is transmitted through the CMG complex or how this energy flow might be influenced by accessory factors such as the FPC. For the same reason, our approach is not suited to address whether strand re-annealing ahead of the fork could reverse conformational transitions and promote backward movement.

To clarify these points, we modified and added the sentences, “**In contrast, Csm3/Tof1, which directly interacts with CMG, may modulate its intrinsic helicase activity in a manner that is independent of DNA sequence. Further investigation will be important to fully elucidate the mechanistic role of Csm3/Tof1 in regulating CMG activity during replication.**” in the Discussion section.